# Increasing tropical cyclone intensity in the western North Pacific partly driven by warming Tibetan Plateau

Jing Xu [1], Ping Zhao [1,2] ✉, Johnny C. L. Chan [3,4], Mingyuan Shi[5], Chi Yang [6], Siyu Zhao [7], Ying Xu[8], Junming Chen[1], Ling Du[9], Jie Wu[10], Jiaxin Ye[1], Rui Xing[11], Huimei Wang[1] & Lu Liu[1]

The increase in intense tropical cyclone (TC) activity across the western North Pacific (WNP) has often been attributed to a warming ocean. However, it is essential to recognize that the tropical WNP region already boasts high temperatures, and a marginal increase in oceanic warmth due to global warming does not exert a significant impact on the potential for TCs to intensify. Here we report that the weakened vertical wind shear is the primary driver behind the escalating trend in TC intensity within the summer monsoon trough of the tropical WNP, while local ocean surface and subsurface thermodynamic factors play a minor role. Through observational diagnoses and numerical simulations, we establish that this weakening of the vertical wind shear is very likely due to the increase in temperature of the Tibetan Plateau. With further warming of the Tibetan Plateau under the Representative Concentration Pathway 4.5 scenario, the projected TCs will likely become stronger.

In recent years, a slew of powerful super typhoons, including Haiyan (2013), Megi (2016), and Mangkhut (2018), has wreaked havoc, causing substantial loss of life and property damage in the coastal countries of East Asia and Southeast Asia. In recent decades, the intense TC, the TC lifetime maximum intensity, and the landfalling TC intensity have increased in the WNP[1–3]. Past and projected increases in human exposure to TCs in many regions[4] heighten the scientific and social concerns further. TC intensity is constrained by large-scale ocean-atmosphere thermodynamic and dynamic factors. High sea surface temperature (SST) and low-level atmospheric moisture may supply more moist static energy for TC genesis and development. Additionally, a weakening of environmental vertical wind shear (VWS) is unfavorable for the intrusion of environmental low-entropy air into the high-entropy reservoir of the TC inner core, which is crucial for maintaining the TC's warm center. Traditionally, it has been widely acknowledged that oceanic thermodynamic factors in the tropical WNP play a more substantial role compared to VWS[2,5–8]. However, it is noteworthy that summer SST in the tropical WNP is already very warm, exhibiting only a feeble or even negative trend in the face of global warming[9–12]. This suggests that the ocean might not supply enough energy for a long-term increase in regional TC intensity[13]. We therefore expect that changes in the atmospheric dynamic factors such as VWS could facilitate TC intensification. While many studies have linked a weakening of VWS to TC intensification[5,14–17], the question of the causes of such a change in VWS has not been investigated specifically.

[1]Chinese Academy of Meteorological Sciences, Beijing 100081, China. [2]Plateau Atmosphere and Environment Key Laboratory of Sichuan Province, College of Atmospheric Science, Chengdu University of Information Technology, Chengdu 610225, China. [3]Guy Carpenter Asia-Pacific Climate Impact Centre, School of Energy and Environment, City University of Hong Kong, Hong Kong, China. [4]Asia-Pacific Typhoon Collaborative Research Center, Shanghai 201306, China. [5]National Meteorological Information Center, Beijing 100081, China. [6]Faculty of Geographical Science, Beijing Normal University, Beijing 100875, China. [7]Department of Atmospheric and Oceanic Sciences, University of California, Los Angeles, Los Angeles, CA, USA. [8]National Climate Center, Beijing 100081, China. [9]Frontier Science Center for Deep Ocean Multispheres and Earth System (FDOMES) and Physical Oceanography Laboratory; and College of Oceanic and Atmospheric Sciences, Ocean University of China, Qingdao 266100, China. [10]Gannan Normal University, Ganzhou 341000, China. [11]Meteorological Service in Binhai New Area, Tianjin 300457, China. ✉e-mail: zhaop@cma.gov.cn

In this work, we show by utilizing observational diagnostics and numerical simulations that the recent warming of the Tibetan Plateau (TP) is the driving force behind a prolonged reduction in VWS within the monsoon trough area of the tropical WNP. The warming TP reinforces the upper-tropospheric South Asian high-pressure system, initiating a meridional wave pattern that extends towards the monsoon trough region. This alters the upper-tropospheric wind, resulting in a reduction of VWS and an increase in TC intensity over the past decades and shedding light on a critical factor in the intensification of these devastating storms. With a further warming of the TP under the Representative Concentration Pathway 4.5 scenario, the stronger TCs are projected. Consequently, the coastal areas of East Asia and Southeast Asia may encounter an increasing risk of severe typhoons.

## Results

### The TC intensity trend in the monsoon trough of tropical WNP

Aircraft-based systematic observations of TCs commenced in the mid-1940s but terminated in the WNP since 1988[18]. The TC intensity estimates based on aircraft measurements are prone to a variety of biases owing to changing instrumentation and means of inferring wind from central pressure[19]. Before the cessation of aircraft reconnaissance, the adjusted TC power dissipation index (PDI) estimate exhibits a strong correlation with SST. However, post-1988, this correlation with SST weakens considerably[19]. A few studies have reported a high correlation between the Atlantic Multidecadal Oscillation (AMO) and the annual

mean TC intensity in the WNP from 1950 to 2018[20]. But this strong correlation has notably diminished after 1987 (see "Methods"). Consequently, the inhomogeneity of TC intensity data due to the halt in aircraft reconnaissance in 1988 has led to an unstable relationship between TC intensity and SST. In this study, we have meticulously selected a relatively homogeneous TC intensity dataset covering the period from 1988 to 2018.

TC activity in the WNP is associated with monsoon trough (MT), monsoon gyres, and monsoon Madden-Julian oscillation[21–23]. The MT indicated by positive vorticity, beginning in July, reaches its strongest level in August and September, and then migrates southward in October (Fig. 1a–f). The MT area, defined by its peak position, spans 12.5°–25° N and 110°–145° E. The mean vorticity over this area increases from $-3.43 \times 10^{-6}$ s$^{-1}$ in June to approximately $3.8 \times 10^{-6}$ s$^{-1}$ in August and September, declining below zero again by November (Fig. 1g). With the strengthening of MT, an anticyclone at 200 hPa emerges near the TP, which signifies the establishment of the South Asian high (SAH). From July to September, the MT is located to the south of the eastern ridge line of SAH. In October, this anticyclone migrates southeastward into the MT area, which indicates the end of the anticyclone over land. Our study focuses on the summer season (July, August, and September; JAS). During this season, TCs are most active and predominantly occur within the MT region in the lower troposphere (Supplementary Fig. 1a), with the prevalence of southwesterly winds in the trough (Fig. 1c–e). Notably, the observed JAS TC day number (50.6 days) represents 52%

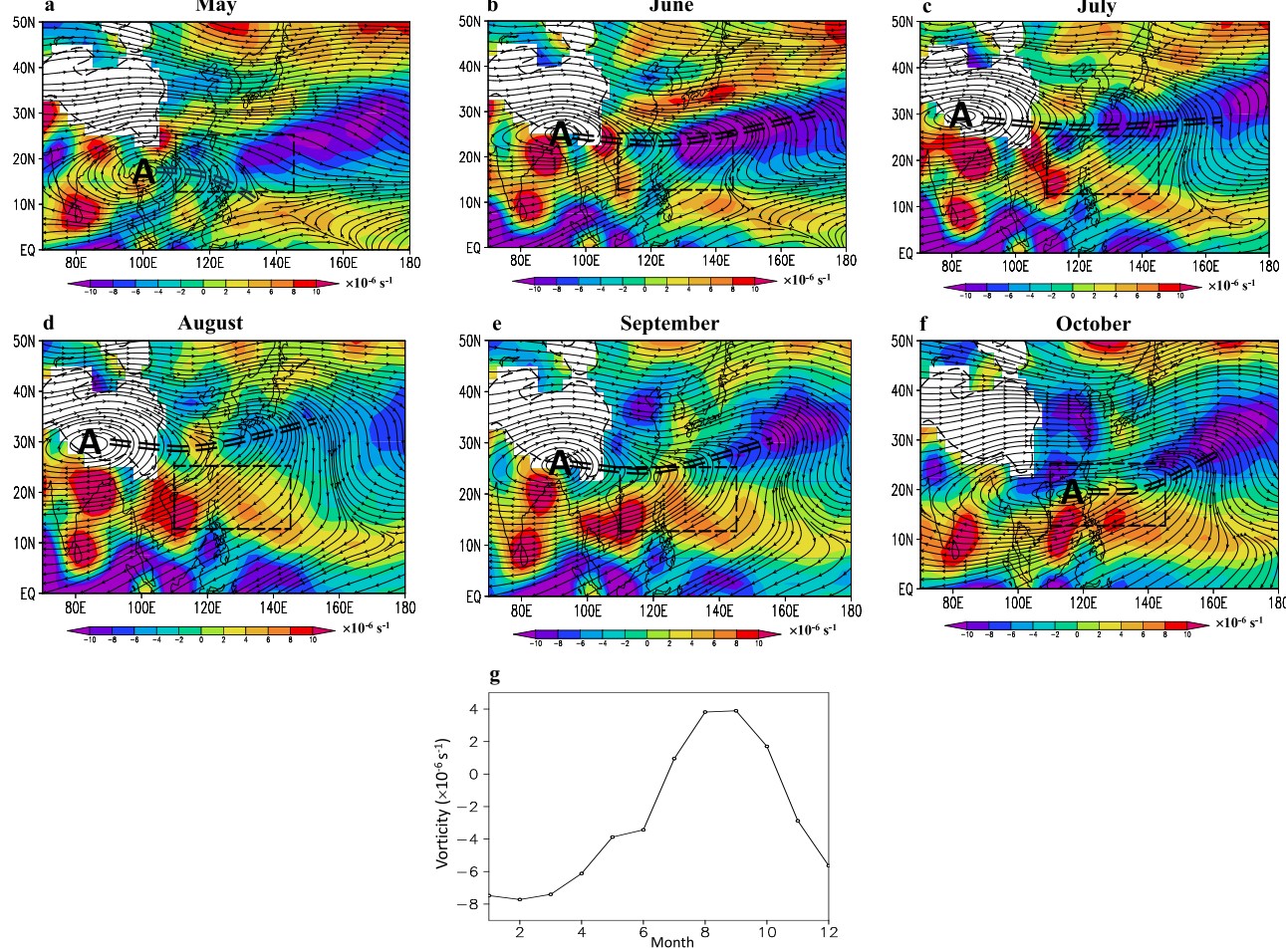

**Fig. 1 | Climatological mean characteristics of observed atmospheric circulation in East Asia-western North Pacific. a–f** 850 hPa vorticity (shaded; × 10⁻⁶ s⁻¹) and 200 hPa flow fields from May to October in turn, in which the black thin dashed line is for topography; the black dashed box is for the monsoon trough (MT) area; "A" indicates the anticyclonic center; and the black double dashed line is for the ridge line of an anticyclone. **g** 850 hPa vorticity ( × 10⁻⁶ s⁻¹) over the MT area. This figure was created using Grid Analysis and Display System (GRADS) Version 2.0.a9.oga.1 (https://sourceforge.net/projects/opengrads/files/). Source data are provided as a Source Data file.

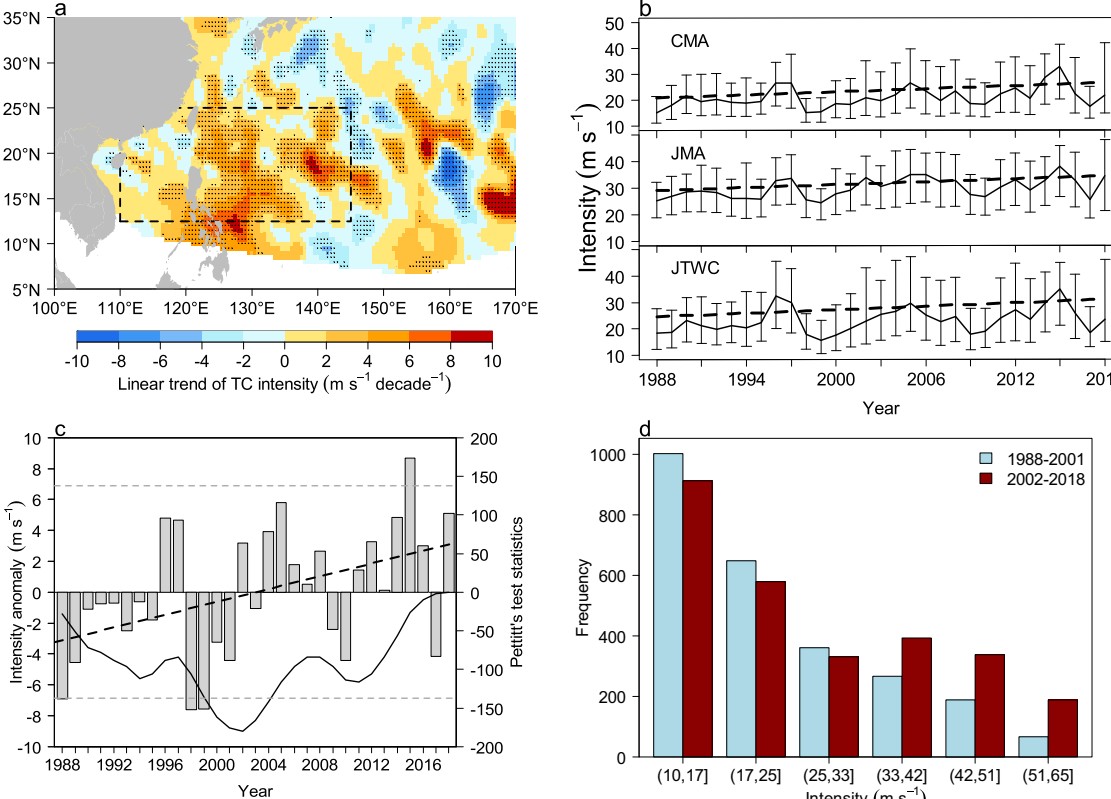

**Fig. 2 | Temporal and spatial variations of observed tropical cyclones (TC) during 1988–2018. a** Linear trend (m s⁻¹ decade⁻¹) of TC intensity, in which dots are statistically significant (at the 0.05 level) and the dashed box indicates the monsoon trough (MT) area. **b** Medians (m s⁻¹; solid lines), interquartile ranges (bar), and linear trends (dashed lines) for TC intensity from the Joint Typhoon Warning Center (JTWC), Japan Meteorological Administration (JMA), and China Meteorological Administration (CMA) datasets in the MT area. **c** The TC intensity anomaly (m s⁻¹; bar) and its linear trend (black long dashed line), the Pettitt's test curve (black solid line) which is used to identify the change point of the TC intensity time series, and the $p = 0.05$ level (short dashed line). **d** Relationship between TC frequency and intensity in the MT area in P1 (period 1988–2001; blue) and P2 (period 2002–2018; red). All data are for July, August, and September only. This figure was created using R version 4.3.2 (https://www.r-project.org/). Source data are provided as a Source Data file.

of the annual TC day number in this area. Given the proximity of the MT area to the coasts of East Asia and Southeast Asia, where hundreds of millions of people reside, it is of special scientific significance to study long-term changes in local TC intensity.

Three TC best-track datasets of IBTrACS[24] from the Joint Typhoon Warning Center (JTWC), Japan Meteorological Administration (JMA), and China Meteorological Administration (CMA) are utilized to examine TC intensity in the WNP. Following ref. 25, the maximum sustained wind (m s⁻¹) is uniformly converted to a 10 min average, in which the correction factors for the JTWC and CMA datasets are 0.88 and 0.87, respectively. Within the MT area, the JAS TC intensity, an average over the JTWC, JMA, and CMA datasets, exhibits an upward trend during 1988–2018 (Fig. 2a). For this figure, geographically-weighted regression (GWR)[26] has been employed to produce the spatial distribution of linear trends of TC intensity (see "Methods"). TC intensity is marked by a significant linear trend of 1–6 m s⁻¹ decade⁻¹. Importantly, this significant upward trend in TC intensity is evident across all three best-track TC datasets, with the mean trends of 2.2 (JTWC), 1.8 (JMA), and 2.0 (CMA) m s⁻¹ decade⁻¹ (statistically significant at the 0.01 level) (Fig. 2b). The maximum trend is observed at the 70th percentile of TC intensity, corresponding to a wind speed of ~31 m s⁻¹ (Supplementary Fig. 1b). This increasing trend is statistically characterized by a change point around 2002 (see "Methods"), with a primarily negative phase during the period 1988–2001 (the earlier period; P1) and a predominantly positive phase during the period 2002–2018 (the later period; P2) (Fig. 2c).

The frequency here refers to the number of TC records with a temporal resolution of 6 h in the TC best-track datasets. The frequency of intense TCs (those with the wind speed > 33 m s⁻¹) exhibits a remarkable increase from P1 to P2 (Fig. 2d). The mean TC intensity in P2 (27.0 m s⁻¹) increases by approximately 19% compared to P1 (22.7 m s⁻¹). Moreover, the numbers of six-hourly records for severe (42–51 m s⁻¹) and super (>51 m s⁻¹) typhoons in P2 stands at 338 and 189, respectively, 1.80 and 2.8 times of the P1 ones (188 and 67 respectively). These results unequivocally indicate the presence of more intense TCs in P2 compared to P1. The TC intensification rate (IR) also shows an increasing trend; and this trend increases with the quantile of IR (Supplementary Fig. 1c), contributing to the overall increase in TC intensity. However, the differences in both latitude and longitude of the TC genesis location within the MT area between P2 and P1 are small (−0.14° and 0.79° respectively, insignificant at the 0.05 level). Similarly, TCs passing through the MT area exhibit negligible shifts in their genesis locations. These analyses imply that the increase in TC intensity in the MT area is not primarily driven by the change of the TC genesis location.

## Links of TC intensity with local oceanic thermodynamic and VWS dynamic factors

Corresponding to the increase in TC intensity, however, positive SST difference between P2 and P1 are generally weak (<0.2 °C) in the tropical WNP. Significant differences in SST are scattered and primarily confined to higher latitudes (Supplementary Fig. 2a). This limited variation in SST in the tropical WNP aligns with prior research

findings[9–12]. In line with a previous study[27], we have applied a 3-year smoothing technique to the TC and other time series (called the smoothed series) to mitigate the impacts of short-time climatic factors. Interestingly, this smoothed TC intensity series exhibits no significant correlation with SST, with a correlation coefficient of 0.18 (Table S1). Ocean heat content (*OHC*) (see Methods) and maximum potential intensity (*MPI*; an integrated metric of the ocean-atmosphere thermodynamic environments in the near-core environment of TC) are often utilized to analyze large-scale environmental factors for the occurrence of TCs. *MPI* may be calculated by SST or the subsurface ocean temperature profile (see Methods). Similar to SST, the smoothed series of *OHC* and *MPI* also do not exhibit significant correlations with TC intensity, having weak correlations of 0.15 and 0.24 respectively with TC intensity in the MT area (Table S1). However, after detrending these series, TC intensity displays high negative correlations with *OHC* (−0.69), SST (−0.58), and *MPI* (−0.5). These insignificant positive correlations or significant negative correlations suggest that the local surface and subsurface oceans might not be capable of supplying the requisite energy to drive the increase in TC intensity. This is distinct from hurricanes in the North Atlantic, where the "storm moisture" supplied by warmer oceans fuels the increase of hurricane intensity in a warming world[27].

VWS is a crucial dynamic factor influencing TC intensity, as evidenced by previous studies[16,28,29]. In contrast to the oceanic thermodynamic factors, VWS (defined as the vertical wind shear vector between 200 hPa and 850 hPa) within the MT area has a high correlation coefficient of −0.54 with TC intensity (significant at the 0.001 level). This correlation is −0.85 (at the 0.01 level) when considering the smoothed series, explaining 72% of variance in the TC intensity. Even in the detrended series, VWS retains a correlation coefficient of −0.48,

significant at the 0.01 level (Table S1). This result underscores a significant connection between VWS and TC intensity. It is evident that TC intensity appears to be more closely linked to VWS than to *MPI* or *OHC* in the MT area. Therefore, we infer that the JAS weakened VWS during the recent three decades is a dominant factor for the increasing trend of TC intensity in the MT area relative to local oceanic thermodynamic factors. Like the previous method[30], such a high correlation can be employed to estimate TC intensity based on VWS.

A closer examination reveals that VWS exhibits a significant reduction of 0.71 m s⁻¹ within the MT area from P1 to P2 (Fig. 3a). This reduction manifests as an anticlockwise vertical wind shear anomaly pattern in the WNP spanning from 12° N to 35° N (Fig. 3b). It is notable that a westerly VWS anomaly is observed in the MT area, which can be attributed to anomalous westerly winds at 200 hPa and anomalous easterly winds at 850 hPa (Fig. 3c–d). Climatologically, the easterly VWS prevails over the tropical WNP during JAS (Supplementary Fig. 3a), with 200 hPa easterly winds ranging from 2 to 8 m s⁻¹ west of 145° E and 850 hPa weaker easterly winds (0 to 3 m s⁻¹) or westerly winds east of 125°E (Supplementary Figs. 3b–c). Consequently, the appearance of westerly VWS anomalies generally signifies a weakening of the 200 hPa easterly winds and a strengthening of the 850 hPa easterly winds, which result in a decrease in the climatological easterly VWS. Within the MT area, VWS exhibits a high correlation with 200 hPa wind, with the correlation of 0.73 (significant at the 0.001 level). This correlation is notably higher than the correlation observed at 850 hPa (0.30). Examining the differences in 200 hPa and 850 hPa winds between P2 and P1 reveals a reduction of 1.23 m s⁻¹ at 200 hPa (at the 0.05 level) and a negligible change of 0.05 m s⁻¹ at 850 hPa. This discrepancy implies a substantial role of the upper-tropospheric circulation.

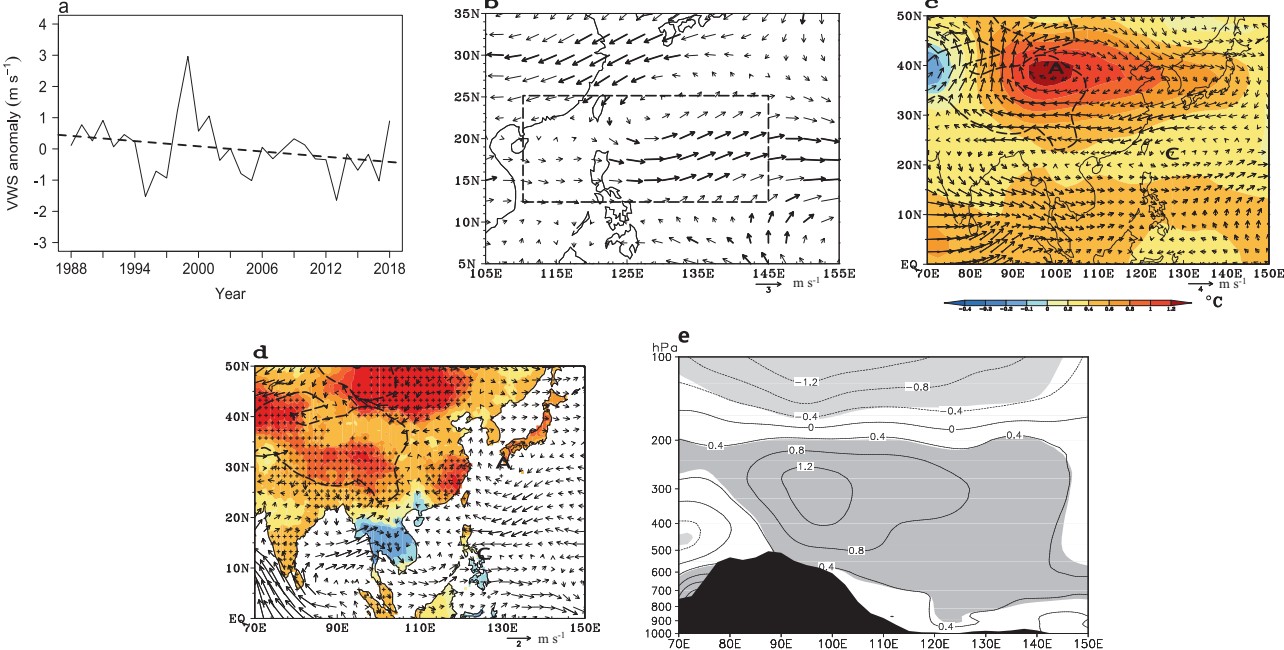

**Fig. 3 | Temporal and spatial variations of observed atmospheric circulation anomalies. a** Time series of the vertical wind shear (VWS) anomaly (m s⁻¹) and its linear trend in the monsoon trough area during 1988–2018. Differences between P2 (period 2002–2018) and P1 (period 1988–2001) in (**b**) VWS between 200 hPa and 850 hPa (m s⁻¹; vector; thick vectors are significant at the 0.05 level). **c** 500–250 hPa mean temperature (°C; shaded; the plus sign is at the 0.05 level) and 200 hPa wind (m s⁻¹; vector). **d** Climate Research Unit (CRU) surface air temperature (°C; shaded; the plus sign is at the 0.05 level) and 850 hPa wind (m s⁻¹; vector), and **e** longitude-height section of temperature (°C) along 32.5°–37.5° N. In **c** and **d**, "A" and "C"

denote the anomalous anticyclonic and cyclonic centers, respectively, and thick vectors are at the 0.05 level; and in **e** light shaded areas are at the 0.05 level and the black shaded area represents topography. Dashed contours in **c–d** represent topography above 1500 m. All data are for July, August, and September only. **a** was created using R version 4.3.2 (https://www.r-project.org/). **b–d** were created using Grid Analysis and Display System (GrADS) Version 2.2.1.oga.1 (https://sourceforge.net/projects/opengrads/files/grads2/2.2.1.oga.1/). **e** Grid Analysis and Display System (GRADS) Version 2.0.a9.oga.1 (https://sourceforge.net/projects/opengrads/files/). Source data are provided as a Source Data file.

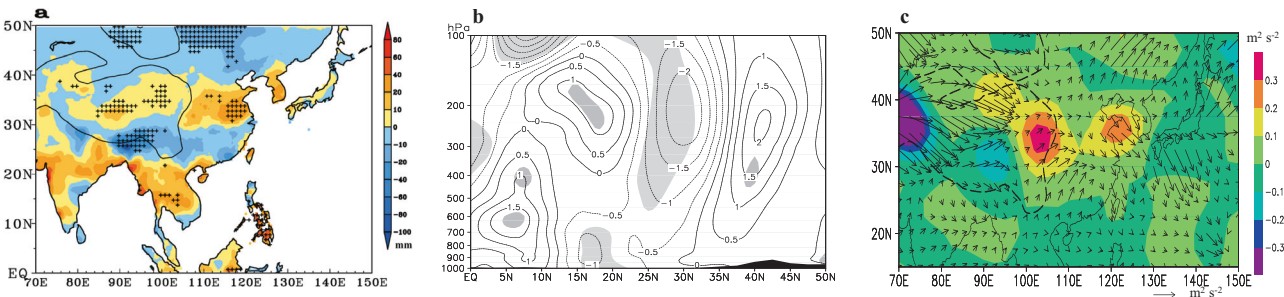

**Fig. 4 | Observed atmospheric circulation and precipitation anomalies between P2 (period 2002–2018) and P1 (period 1988–2001). a** Difference in Climate Research Unit precipitation (mm; the plus sign is at the 0.05 level). **b** Latitude-height section of differences in zonal wind (m s⁻¹; light shaded areas are at the 0.05 level) along 127.5° E. **c** 500–200 hPa $\mathbf{F_h}$ (m² s⁻²; vector) and $F_z$ (m² s⁻²; shaded) of anomalous wave. All data are for July, August, and September only. **a** was created

using Grid Analysis and Display System (GrADS) Version 2.2.1.oga.1 (https://sourceforge.net/projects/opengrads/files/grads2/2.2.1.oga.1/). **b–c** were created using Grid Analysis and Display System (GRADS) Version 2.0.a9.oga.1 (https://sourceforge.net/projects/opengrads/files/). Source data are provided as a Source Data file.

---

Thus, the shift in the upper-tropospheric circulation has a larger contribution to the weakened VWS in the MT area. The mechanism driving this change in the upper-tropospheric circulation within the MT area warrants further investigation.

### The impact of warming TP

Climatologically, the SAH dominates the mid-low latitudes of the Asian continent and extends into the WNP (Supplementary Fig. 3b). The MT area lies in the southeastern part of the SAH and is characterized by predominantly easterly winds, which constitute the primary component of VWS in this area. An extensive anticyclonic anomaly emerges in extratropical East Asia at 200 hPa, featuring positive anomalies in geopotential height expanding both upward and eastward from the TP, with an anomalous center in the northeastern TP (Fig. 3c, Supplementary Fig. 3d–e). This pattern signifies an eastward strengthening of the SAH. The smoothed surface pressure in the central-western TP (80°–95° E, 29°–34° N) exhibits a significant correlation of 0.64 with TC intensity in the MT area (Supplementary Table 1), which suggests that the long-term change in TC intensity may be modified by the TP signal. The eastward expansion of the SAH can be explained as follows. Corresponding to a warming TP, tropospheric temperature increases locally, westerly wind north of the SAH center might intensify the transport of warmer air toward the east, leading to positive temperature advection ($-\mathbf{V} \cdot \nabla T$) in the troposphere east of the SAH center (Supplementary Fig. 4a) and increases in temperature in this area. Concurrently, the absolute vorticity advection decreases with height in the region east of the strengthened SAH center (Supplementary Fig. 4b). To satisfy the vorticity equation[31] (see "Methods"), anomalous upward motion typically emerges in the troposphere east of the SAH center (Supplementary Fig. 4c), which causes the increased precipitation over East Asia-north of 30° N (Fig. 4a). The condensational heating feedback generated by precipitation (Supplementary Fig. 4d) as well as temperature advection (shown in Supplementary Fig. 4a) might locally warm the troposphere and cause the observed eastward extension of the warming troposphere, which elongates the tropospheric air column and subsequently raises isobaric surfaces in the upper troposphere. As a result, a local anticyclonic anomaly emerges in the upper troposphere, thereby causing the eastward extension of the SAH. This process results in a widespread increase in geopotential height over the extratropical regions of East Asia and the WNP (Supplementary Fig. 3e).

The westerly and easterly wind anomalies generally prevail north and south of the anomalous SAH center, respectively, and a meridional anomalous wave train in zonal wind emerges in the upper troposphere over the WNP, with the westerly wind anomalies around 200 hPa in the MT area (Fig. 4b). An anomalous cyclone between the subtropical

easterly wind anomalies and the tropical westerly wind ones in the middle-upper troposphere covers a large region of the WNP (Fig. 3c). This change in upper-tropospheric wind over the MT area contributes to the emergence of westerly VWS anomalies. The meridional wave train observed in the WNP is associated with a southward transport of wave activity flux from East Asia. Our analysis shows large horizontal vectors ($\mathbf{F_h}$; see "Methods") of wave activity flux from the eastern TP into the extratropical WNP (Fig. 4c), where the anticyclonic anomaly is located. These vectors expand southward into the tropical WNP and may intensify local wave activities[32]. Concurrently, there are large positive $F_z$ in East Asia-north of 30°N, with the largest positive center located in the eastern TP, which strengthens the upward transport of wave activity flux. In the framework of the wave activity[32], thermal forcing might be an important contributor to the generation of wave activities.

Indeed, the anticyclonic anomaly centered in the TP is accompanied by a local warming in the middle to upper troposphere (500–250 hPa), with a temperature increase of 1.2 °C in the northeastern TP (Fig. 3c). Vertically, this warming extends from the surface of the TP to the upper troposphere over East Asia and the WNP (Fig. 3e). Surface air temperature (SAT) also increases in the TP area (Fig. 3d, Supplementary Fig. 3f). In particular, the regional mean value of the CRU SAT exhibits a significant increase of 0.50 °C (at the 0.001 level) (0.56 °C at meteorological stations) in the TP area (with topography > 1500 m) during P2 relative to P1, which underscores a remarkable warming phenomenon in the TP area.

The above evidence reveals that corresponding to the upward trend of the TC intensity, the TP is warming up and the SAH strengthens, with the upper-tropospheric cyclonic anomaly in the subtropical WNP, which weakens the VWS in the MT area of the tropical WNP. It is therefore hypothesized that these changes in atmospheric circulation from East Asia to the tropical WNP can be forced by a warming TP.

To explore this hypothesis, we simulate the impact of the TP elevated surface heating which more greatly modifies the SAH and atmospheric circulation in the North Pacific compared to the plain areas of the Asian continent[33–36]. For this purpose, the National Center for Atmospheric Research (NCAR) Community Climate System Model, version 3 (CCSM3), an ocean-atmosphere coupled model, is conducted through changing the TP surface albedos (see "Methods"). Two experiments (CCSM3_TP_Veg and CCSM3_TP_Soil) are designed. Although this uniform modification in the TP area possibly causes a bias between the observed and modeled heat distributions, the model still captures the primarily characteristics of the observational atmospheric circulation changes.

Relative to CCSM3_TP_Soil (a cool TP surface), CCSM3_TP_Veg (a warm TP surface) forces a warming troposphere in the TP area which

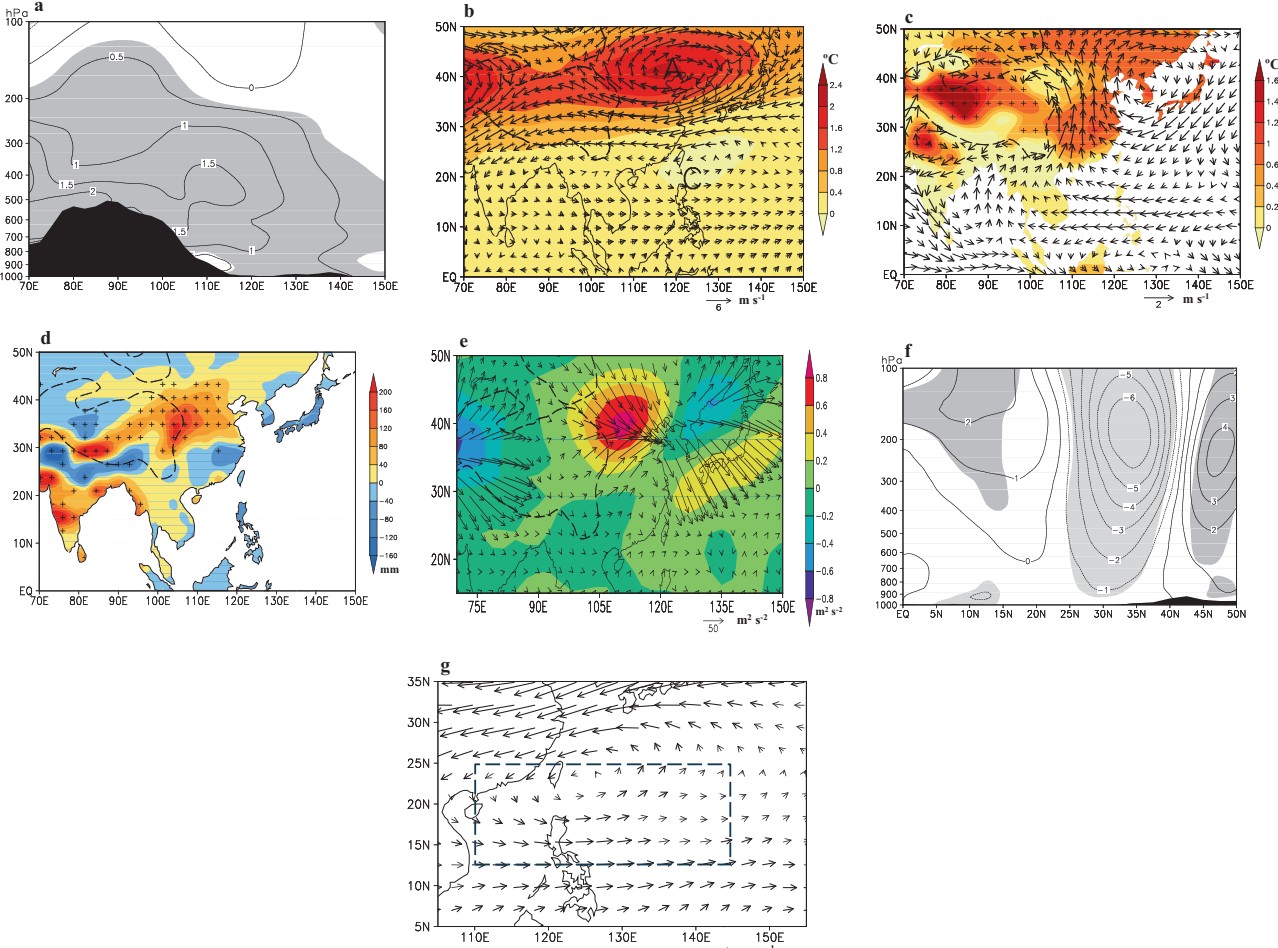

**Fig. 5 | Anomalies forced by the warming Tibetan Plateau (TP). a** For the longitude-height section of temperature (°C) along 32.5°–37.5° N, in which light shaded areas are at the 0.05 level. **b** For 500–250 hPa mean temperature (°C; shaded; the black plus sign is at the 0.05 level) and 200 hPa wind (m s⁻¹; vector; thick vectors are at the 0.05 level). **c** For surface air temperature (°C; shaded; the black plus sign is at the 0.05 level) and 850 hPa wind (m s⁻¹; vector; thick vectors are at the 0.05 level). **d** For precipitation (mm; the black plus sign is at the 0.05 level). **e** For 500–200 hPa $F_h$ (m² s⁻²; vector) and $F_z$ (m² s⁻²; shaded). **f** For latitude-height section of zonal wind (m s⁻¹; light shaded areas are at the 0.05 level) along 127.5° E. **g** For vertical wind shear (m s⁻¹; thick vectors are at the 0.05 level) between 200 hPa and 850 hPa. All data are for July, August, and September only. This figure was created using Grid Analysis and Display System (GRADS) Version 2.0.a9.oga.1 (https://sourceforge.net/projects/opengrads/files/). Source data are provided as a Source Data file.

expands into extratropical East Asia, and forces a warming East Asian continent (Fig. 5a–c). Like the observation, the simulated positive temperature advection anomalies emerge in the troposphere over extratropical East Asia (Supplementary Fig. 5a) and absolute vorticity advection decreases with height in this region (Supplementary Fig. 5b). There is an anticyclonic anomaly in extratropical East Asia and a cyclonic anomaly in the subtropical WNP in the upper troposphere. Upward motion and precipitation increase in East Asia-north of 30°N (Fig. 5d, Supplementary Fig. 5c), which is consistent with the result forced by the warming TP in an atmospheric general circulation model with a climatological mean SST[37]. This phenomenon implies a minor effect of the ocean-atmosphere interaction on precipitation in extratropical East Asia. Meanwhile, the TP warming also successfully simulates the propagation of wave activity flux from East Asia to the tropical WNP, the meridional wave train in the WNP, and the anticlockwise VWS anomaly with the westerly VWS anomaly in the MT area (Fig. 5e–g). It is evident that the CCSM3 model captures a response similar to the observational changes associated with the increase in TC intensity in the MT area (though there are slightly northward locations in simulation compared with the observation). The similarity between model and observations offers evidence that our results are not model-dependent. Therefore, the observed atmospheric conditions favorable

to TC development could be physically generated by the warming TP. In addition, the TP topographic dynamic roles may also affect the downstream atmospheric circulation through Rossby waves[38].

Our results strongly suggest that the increasing trend of JAS TC intensity in the MT area of the tropical WNP is closely tied to a warming TP. This warming intensifies the SAH, triggering atmospheric circulation anomalies over the WNP, contributing to a reduction in environmental VWS and TC intensification within the MT area (Fig. 6).

## A comparison with Eurasian land and global SST forcings

A significant warming is also observed in other areas of Asia and Europe (Supplementary Fig. 6a). To compare with the impact of TP, we conduct sensitivity experiments for Asian or European warming (see Methods). The results indicate that warming in Asia does induce an anticlockwise VWS anomaly in the tropical WNP. However, this anomaly is situated father south, and the westerly VWS anomalies are confined to a narrower zonal belt (south of 16°N) (Supplementary Fig. 6b). These anomalies are weaker in the MT area when compared to the effects of TP warming, which reduces the impact of the warming TP on VWS in the MT area. This weakening of VWS driven by warming in Asia primarily results from lower-tropospheric westerly wind anomalies in the MT area (Supplementary Figs. 6c–d), which differ from the

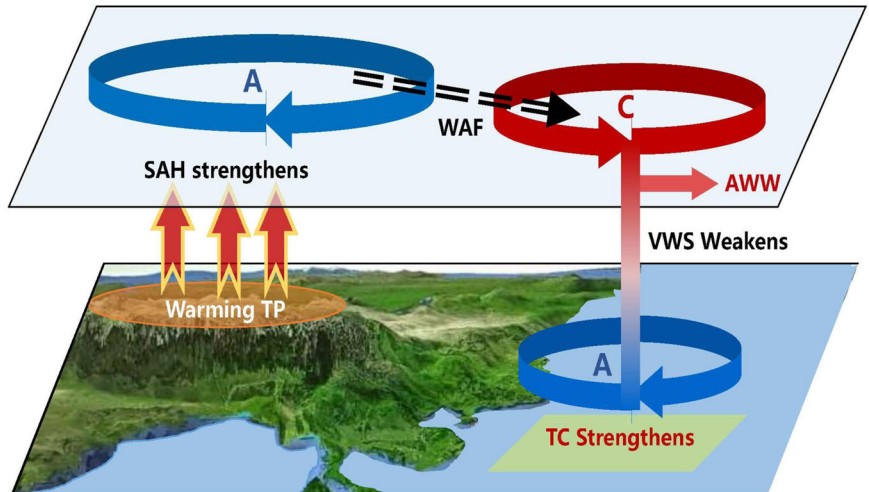

**Fig. 6 | Schematic diagram of the impact of Tibetan Plateau (TP) warming on tropical cyclones (TC) intensity in the monsoon trough (MT) area.** Warming TP strengthens the South Asian high (SAH), triggering the southwestward propagating wave activity flux (WAF) and the high-level anomalous cyclone and westerly wind in the western North Pacific, weakening the vertical wind shear (VWS) in the MT area, and strengthening TC intensity. A: anomalous anticyclonic center; C: anomalous cyclonic center; and AWW: anomalous westerly wind.

observed and TP-induced easterly wind anomalies. Relative to warming in Asia, warming in Europe forces a weaker VWS in the MT area (Supplementary Figs. 6e–g). In fact, the observed warming pattern in Eurasia might also be attributed to the influence of the TP warming (Supplementary Fig. 5d). This impact of the TP on increasing European temperatures during the summer months has been documented in the previous studies[39]. The above analysis supports the independent impact of the warming TP on VWS in the MT area.

One might speculate that the global pattern of SST change from P1 to P2 plays the dominant role in the weakened VWS over the MT area. To investigate this potential impact, we perform two sensitivity experiments using the NCAR CAM5 climate model with prescribed SST (see "Methods"). The results show that the observed global distribution of SST change between P2 and P1 (Supplementary Fig. 2b) does not generate a pronounced anticlockwise VWS anomaly or the westerly VWS anomaly observed in the MT area (Supplementary Fig. 2c), unlike the observed (Fig. 3b) and the TP-induced (Fig. 5g). This finding excludes the dominant role of SST in the reduction of VWS from P1 to P2. Furthermore, it is worth noting that when the atmospheric anomalies in the Asia-North Pacific region are induced by TP warming, significant positive SST anomalies emerge in the North Pacific, with relatively weak SST anomalies in the MT area (Supplementary Fig. 5e), which is similar to the observation. This suggests that a warming TP can induce the SST anomalies observed in the WNP, consistent with the previous research[36,40]. These studies have demonstrated the modeled impact of TP heating anomalies on the Pacific SST and mixed layer, shedding light on the significance of TP heating for the Asian–Pacific climate.

Our results strongly suggest that relative to a substantial role of TP warming in reducing VWS and increasing TC intensity in the summer MT area of the tropical WNP over the past three decades, both the global SST change and warming in other regions of Eurasia have had minor roles during this period.

## Discussion

Temperature records based on tree rings for the past 2485 years on the central-eastern TP reveal a long-term warming trend commencing in ~1600 AD[41]. This warming was accelerated over the past decades, showing a larger magnitude during 1991–2020 (approximately our study period) than during 1961–1990[42]. Based on the optimal fingerprinting attribution method and the model simulations of the Coupled Model Intercomparison Project Phase 5 (CMIP5) and the Coupled Model Intercomparison Project Phase 6 (CMIP6), previously studies consistently find that the recently accelerated warming in the TP is very likely dominated by anthropogenic (ANT) influence[43–45]. The trends of SAT during 1985–2014 in all seasons are positive over the TP, and the observed warming amplification during 1961–2014 has been attributed to ANT influence[45]. The attributable contribution from ANT is estimated to be much larger than that from natural signal for most warming events in the TP[46]. Under the Representative Concentration Pathway (RCP) 4.5 scenario, a medium emission scenario, the CMIP5 multi-model ensemble projects a continued decrease in TP snow cover and ongoing warming of the TP[47].

While the oceanic conditions in the tropical WNP may not inherently support the increases in TC intensity in the future scenario[48], in accordance with our finding, the persistent warming of the TP is expected to contribute to a reduction in VWS. This, in turn, favors an increase of TC intensity within the MT area (Supplementary Fig. 7a). Given the high correlation between TC intensity and VWS in this area, the projected TC intensity is anticipated to rise in the future (Supplementary Fig. 7b). Similar to the CCSM3 model, the CMIP5 HadGEM2-ES global circulation model adequately replicates the warming of the TP and the westerly VWS anomaly within the MT area of the tropical WNP (Supplementary Fig. 8b). Downscaling projections with the high-resolution RegCM4 model (see "Methods") supports this increase in TC intensity in the future scenario relative to the historical simulations, with a significant increase of 2.4 m s⁻¹ (an increase of 8.2%, significant at the 0.001 level) in TC intensity and a notable increase in the number of intense TCs (Supplementary Figs. 8c–d). These projections suggest a more frequent occurrence of intense typhoons in the region in the coming decades, which heightens the risk of powerful typhoons striking the coastlines of East Asia and Southeast Asia, including Philippines and the southern mainland of China.

It is noted that due to regional and seasonal differences in the Asian–Pacific climate background, this mechanism does not negate the potential influence of other oceanic and atmospheric factors on VWS and TC development in other regions and seasons. Furthermore, in the MT area, the detrended series of oceanic thermodynamic factors such as *OHC*, SST, and *MPI* consistently exhibit strong negatively correlations with that of TC intensity, which differs remarkably from the traditional findings that had documented significant positive correlations between these variables and also poses a challenge to the

relationship between TC and ocean. These topics remain subjects of ongoing investigation.

## Methods

### Observation, reanalysis, and projection datasets

This study utilizes three TC best-track datasets of IBTrACS[24] from JTWC, JMA, and CMA, spanning the years 1988–2018. SAT data at 2209 meteorological stations, marked by red and blue dots in Supplementary Fig. 3f, are collected from the National Meteorological Information Center of China. Additionally, monthly SAT and precipitation data with a horizontal resolution of 0.5° in both latitude and longitude from the CRU during 1988–2018[49], as well as monthly Hadley Centre SST with a horizontal resolution of 1° in both latitude and longitude during this period[50], are incorporated into the analysis. Large-scale atmospheric conditions were examined using the NCEP/NCAR atmospheric reanalysis monthly data with a horizontal resolution of 2.5° in both latitude and longitude[51]. Monthly subsurface temperature profiles in the ECMWF Ocean Reanalysis System 5 (ORAS5) dataset[52], with a horizontal resolution of 1° in both latitude and longitude and 75 levels from the surface to 6000 m, are utilized to calculate *OHC*. In addition, air temperature, wind, and surface temperature with 288 and 192 grid points respectively in longitude and latitude from the CMIP5 CCSM4 historical (1986–2005) simulation and future projection (under the RCP4.5 emission scenario)[53] are also utilized in this study.

### Analysis for the inhomogeneity of TC intensity time series around the termination of aircraft reconnaissance

Prior to the availability of satellite observations, the determination of TC intensity relied on limited in situ data[19]. Aircraft systematic observations of TCs commenced in the mid-1940s but terminated in the WNP region since 1988[18]. TC intensity estimates derived from aircraft measurements are prone to a variety of biases due to changing instrumentation and methods of inferring wind from central pressure[19]. All these might lead to the inhomogeneity of the observational intensity time series over the WNP even though the global satellite observation was routinely available before 1988. Consequently, the TC intensity trends reported in JTWC, JMA, and CMA datasets remain contentious[8,19,54,55].

For instance, prior to the cessation of aircraft reconnaissance, the adjusted TC PDI estimate from JTWC closely align with the unadjusted estimate from JMA, and both correlated well with SST. However, after the termination of aircraft reconnaissance, the estimates diverge significantly, and their correlations with SST weaken[19]. In addition, several studies have reported a strong correlation (0.86) between the Atlantic Multidecadal Oscillation (AMO) and the annual mean WNP TC intensity during 1950–2018[20]. To explicitly understand this correlation, we repeat their analysis using the JTWC best-track annual TC data from 1950–2018. This high correlation is observed mainly before the late 1980s, with this in-phase relationship diminishing significantly thereafter. The 30-year running correlation notably declines after 1987 and reaches its lowest values during 1988–2017 and 1989–2018, with correlations of only 0.04 and 0.03, respectively. This underscores that the inhomogeneity caused by measurement changes can contribute to an unstable relationship between SST and TC intensity. It is also noted that JTWC adjusted their wind-pressure relationship for TC intensity estimation around 2007. However, TC intensity series from JTWC, JMA, and CMA shows the high consistency around 2007. Since this change in wind-pressure relationship did not occur in the CMA data, the consistency between the JTWC and CMA datasets suggests that the modification in the JTWC wind-pressure relationship had a little effect on the inhomogeneity of JTWC TC intensity data around 2007.

### Statistical analysis

This study employs least squares trend and composite analysis methods to assess the linear trend of TC intensity and the differences between the two fields, respectively. The statistical significance of composite differences and nonzero trends is determined using the Student's *t*-test. The Pettitt's test[56] is applied to identify abrupt change points in the three-year running mean TC intensity series, revealing a significant change point in the time series around 2002. We use the "Random-phase" test to assess significance for three-year running mean time series. All reported significances are at the *p* = 0.05 level unless stated otherwise.

The GWR[26] is utilized to generate spatial distributions of linear trends of TC intensity. The GWR model for spatially-varying linear trends is as follows.

$$y_i = \beta_0(u_i, v_i) + \beta_1(u_i, v_i)x_i + \varepsilon_i, \varepsilon_i \sim N(0, \sigma^2) \quad (1)$$

where $u_i$ and $v_i$ are the longitude and latitude of TC intensity $y_i$, respectively; $x_i$ is the year of $y_i$; $\beta_0$ is the interception term; and $\beta_1$ is the linear trend. $\beta_0$ and $\beta_1$ are continuous functions of longitude and latitude, estimated by a weighted least squares method with a kernel function for the weight estimation. The fitted function $\beta_1$ is projected onto a 0.5°latitude × 0.5°longitude grid only for plotting the linear trend.

Following the previous method[30], capitalizing on the high correlation coefficient between TC intensity and VWS, we have employed the smoothed series to establish a physical-empirical model as follows.

$$I_{TC} = -7.197 \times I_{VWS} + 106.025. \quad (2)$$

Here $I_{TC}$ represents the predicted TC intensity, and $I_{VWS}$ denotes VWS in the MT area. This equation has been statistically validated with the *F*-test, which is found to be significant at the *p* = 0.001 level. The Root Mean Square Error (RMSE) between the simulated and observed $I_{TC}$ series is 1.38 m s⁻¹, constituting 6% of the observed $I_{TC}$ series.

### Oceanic thermodynamic diagnosis

The ocean heat content (*OHC*) calculation is based on the temperature analysis above the 26 °C isotherm depth[7]. Grid-wise *OHC* is defined as a vertical integration of heating profile, calculated as follows.

$$OHC = \sum_{z_0}^{z_{26}} c_{p_i} \rho_i (T_i - 26) \Delta z_i \quad (3)$$

Here $\rho_i$ is the seawater density, $C_{pi}$ is the specific heat of seawater, and $Z_{26}$ is the depth 26 °C isotherm in each grid. *OHC* is calculated with the ORAS5 monthly temperature profile above the 26 °C isotherm depth.

In this study, the maximum potential intensity (*MPI*), which reflects the integrated influence of the ocean-atmosphere thermodynamic conditions in the near-core environment of TC[57], is expressed as follows.

$$MPI = \alpha \sqrt{\frac{T_s}{T_0} \frac{C_k}{C_D} \left[ CAPE^* - CAPE \right]\big|_{RMW}} \quad (4)$$

Here $C_k$ is the surface exchange coefficient, $C_D$ the surface drag coefficient, $T_s$ the SST, and $T_o$ the outflow-layer air temperature. $T_s/T_o$ indicates an effect of dissipative heating. *CAPE** is the convective available potential energy of air lifted from saturation at sea level through the environmental sounding, while *CAPE* represents that of air in the boundary layer. *CAPE** and *CAPE* are calculated near the radius of maximum wind (*RMW*) of the hypothetical cyclone using a reversible adiabatic parcel-lifting algorithm. Consistent with ref. 57, the reduction factor of gradient wind to surface wind (α) is taken as 0.8, and the ratio of $C_k$ to $C_D$ is assumed to be 0.9. Input data include SST, surface pressure, and the vertical profiles of the atmospheric temperature and humidity The source code is available at ftp://texmex.mit.edu/pub/

emanuel/TCMAX/. In addition, MPI may also be calculated by the subsurface ocean temperature profile, which can more realistically characterize the ocean contribution to TC intensity[6].

### Atmospheric thermodynamic and dynamic diagnosis

In accordance with the standard vorticity equation, the following expression can be derived[31]

$$\frac{\partial^2 w}{\partial z^2} = f^{-1}\frac{\partial}{\partial z}(\zeta'_t + \mathbf{V}\cdot\nabla\zeta + \beta v) \tag{5}$$

Here $w$ is the vertical velocity; $\mathbf{V}$ is the horizontal wind; and $\zeta$ is the relative vorticity. We calculate advection of relative vorticity ($\mathbf{V}\cdot\nabla\zeta$) and advection of planetary vorticity ($\beta v$). Using a normal mode solution, $w$ can be expressed as follows[31].

$$w \propto -f^{-1}\frac{\partial}{\partial z}(\mathbf{V}\cdot\nabla\zeta + \beta v) \tag{6}$$

When $\frac{\partial}{\partial z}(\mathbf{V}\cdot\nabla\zeta + \beta v) < 0, w > 0$.

The apparent moisture sink $Q_2$ is written as follows[58].

$$Q_2 = -L\left(\frac{\partial q}{\partial t} + \mathbf{V}\cdot\nabla q + \omega\frac{\partial q}{\partial p}\right) \tag{7}$$

Here $q$ is the mixing ratio; $\omega$ is the $p$-vertical velocity; and $L$ is the latent heat of condensation. $<Q_2> = \frac{1}{g}\int_{p_T}^{p_S} Q_2 dp$.

Wave activity flux (WAF) serves as a tool to analyze the activity area of anomalous wave and the wave propagation, aiding in the identification of the origin of a wave train. We adopt the conventional definition of WAF ($\mathbf{F}_s$) as outlined below[32].

$$\mathbf{F}_s = p \times \cos(\varphi) \times \begin{pmatrix} F_x \\ F_y \\ F_z \end{pmatrix} \tag{8}$$

In this equation, $F_x$, $F_y$, and $F_z$ represent the zonal, meridional, and vertical components of $\mathbf{F}_s$, respectively. All parameters in the above equation align with those from ref. 32. The horizontal vector of $\mathbf{F}_s$ is defined as $\mathbf{F}_h = (F_x, F_y)$.

### Simulations of the CCSM3 and CAM5 models

The original NCAR CCSM3 with a horizontal resolution of 2.8° in latitude and longitude and 26 vertical levels[59] encompasses an atmospheric component, an oceanic component, a land model, and a sea ice model. This model is employed to investigate the impact of a warming TP on the atmospheric circulations in East Asia and the tropical WNP. Previous studies have validated the CCSM3's ability to simulate the effects of Asian land heating on the Asian–Pacific climate[35,40]. For this study, three sensitivity experiments concerning the TP are conducted: CCSM3_CTL, CCSM3_TP_Veg and CCSM3_TP_Soil. CCSM3_CTL employs the original model. Referring to previous studies[34,35,37], in CCSM3_TP_Veg, the surface vegetation type at each grid point of the TP area with topography > 1500 m is set to needleleaf evergreen temperate tree, which has a lower surface albedo. CCSM3_TP_Soil mirrors CCSM3_TP_Veg but replaces vegetation with bare soil, which has a higher surface albedo. In these experiments, CCSM3 is integrated for 50 years, with analysis focused on the outputs from the final 20 years. This setup is equivalent to the results from an ensemble of 20 sensitivity experiments with changes in initial atmospheric and land surface conditions[35,60]. This artificial reduction in albedo on the TP results in a local SAT increase, which mimics a TP warming scenario[34]. Although this uniform modification in vegetation of the TP region possibly introduces a bias between the observed and modeled heat

distributions, the model successfully captures the primarily characteristics of the observed atmospheric circulation changes. Similarly, the sensitivity experiments are designed for warming Asia (60°–120° E, 25°–50° N) and warming Europe (0°–60° E, 25°–50° N).

The original NCAR CAM5 model with a horizontal resolution of 2.5° × 1.89° in longitude and latitude[61] and prescribed SST is employed to investigate the impact of the global SST change from P1 to P2. Two sensitivity experiments are designed: CAM5_SST_P1 and CAM5_SST_P2. The observed JAS global SST distribution averaged over 1988–2001 and 2002–2018 are used in CAM5_SST_P1 and CAM5_SST_P2, respectively. In these experiments, CAM5 is integrated over 20 years, equivalent to the results from an ensemble of 20 sensitivity experiments with changes in initial atmospheric and land surface conditions.

### Downscaling TC intensity with the high-resolution RegCM4 model

The RegCM4 climate model with a horizontal resolution of 25 km, 18 vertical sigma layers, and the atmospheric top at 10 hPa was developed by the Abdus Salam International Center for Theoretical Physics (ICTP). This model has been employed as part of Phase II of the CORDEX-East Asia domain (with a region 72°–157° E, 13° N-48° N) (Supplementary Fig. 8a) to simulate the climate characteristics in East Asia and the WNP, including long-term trends in the WNP TC intensity[62–64]. In these simulations, initial and evolving lateral boundary conditions, including atmospheric temperature, specific humidity, and wind, surface pressure, and SST, are applied to drive the RegCM4 model. They are interpolated from the CMIP5 HadGEM2-ES global circulation model outputs[65]. The lateral boundary conditions and the prescribed SST are updated every 6 h. The HadGEM2-ES has demonstrated the ability to reasonably simulate atmospheric circulation in East Asia and the Western Pacific[66]. The RegCM4 simulations span two time period: 1968–2005 with observed greenhouse gas (GHG) concentrations and 2006–2098 for the future under the RCP4.5 scenario. GHG concentrations are updated each year throughout the simulation period. We utilize the RegCM4 output under the RCP4.5 scenario[63,64]. In line with ref. 64, the reference period 1986–2005 is selected.

To detect and track TCs in simulation, the TSTORMS (Detection and Diagnosis of Tropical Storms in High-Resolution Atmospheric Models) software (www.gfdl.noaa.gov/tstorms/) is utilized. In this software, TCs are identified based on criteria including the 850 hPa relative vorticity maximum, local sea level pressure minimum, warm core center, and SST threshold. Trajectory analysis is conducted to link TC centers in time for a given track. This considers the closest TC within 400 km over a 6 h period, with preference given to TCs located westward and poleward of the current position, if multiple candidates existed. In addition, following ref. 64, we calculate the basin-, time-, and model-dependent vorticity as proposed by ref. 67. A vorticity threshold is defined as twice the standard deviation of the 850 hPa relative vorticity in the study area. A TC trajectory is considered valid if the TC lasted at least one day. Similar to with previous reports[68,69], our simulation likely underestimates the number of TCs due to the coarse horizontal resolution. Since this study is interested in the long-term change in TCs, the TC number issue might become less critical as it affects both present and future cyclones.

### Data availability

The IBTrACS (version 4) of TCs are available online https://www.ncei.noaa.gov/products/international-best-track-archive. The CRU monthly surface air temperature and precipitation are obtained from https://crudata.uea.ac.uk/cru/data/hrg/. The Hadley Centre SST data are available at https://www.metoffice.gov.uk/hadobs/hadisst/. Surface air temperature data at meteorological stations are from https://data.

cma.cn/. The subsurface temperature profile dataset in the ECMWF Ocean Reanalysis System 5 dataset is available at https://icdc.cen.uni-hamburg.de/thredds/catalog/ftpthredds/EASYInit/oras5/catalog.html. The NCEP monthly mean reanalysis dataset is obtained online from the home page (https://psl.noaa.gov/data/reanalysis/reanalysis.shtml). The CMIP5 data are obtained at https://esgf-node.llnl.gov/search/cmip5/. Source data for figures and tables are provided with this paper. Extended data in this study are available at https://figshare.com/articles/dataset/CCSM3_model_data/24585597, https://figshare.com/articles/dataset/CAM5_model_data/24581193, and. https://figshare.com/articles/figure/RegCM4_TC_information/24585456. Source data are provided with this paper.

## Code availability

All of the figures are created by the authors using the NCAR Command Language Version 6.5.0 or 6.3.0 (available at http://www.ncl.ucar.edu/), or Grid Analysis and Display System (GRADS) (http://cola.gmu.edu/grads/grads.php), or R package (https://www.r-project.org/). The code for MPI is available at ftp:texmex.mit.edu/pub/emanuel/TCMAX/ and the TSTORMS software is available at www.gfdl.noaa.gov/tstorms/. Other analytical scripts are available from the authors upon request.

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

## Acknowledgements

This work is jointly sponsored by the Second Scientific Expedition on the Qinghai-Tibet Plateau (2019QZKK020803), the National Key R&D Program of China under grant 2022YFC3004200, and Key Special Projects of National Key R&D Program of China (2018YFC1505700).

## Author contributions

J.X., C.Y., L.L., and R.X. conducted the analyses on TC changes and atmospheric circulation features with observation datasets, and CMIP5 CCSM4 projection datasets and wrote some sections of the manuscript; P.Z. conceived the study framework, devised the study schemes, analyzed atmospheric circulation features with observation and CCSM3 and CMA5 model experiment datasets, and wrote and revised the manuscript; C.J.C.L. discussed the results and revised the manuscript; M.Y.S. and J.M.C. conducted the CCSM3 model experiments; J.X.Y. conducted the CAM5 model experiments; S.Y.Z. dynamically analyzed the physical mechanism responsible for the effect of the warming TP on atmospheric circulation in the tropical Pacific with the reanalysis dataset and revised the manuscript; Y.X. and J.W. conducted and analyzed the projected results with the RegCM4 model and CMIP5 HadGEM2-ES model outputs; L.D. conducted the calculation and analysis on ocean heat content; and HMW took part in the data treatment.

## Competing interests

The authors declare no competing interests.
