## [Peer Review File · Nature Communications]

Increasing tropical cyclone intensity in the western North Pacific partly driven by warming Tibetan PlateauReviewers' comments:

Reviewer #1 (Remarks to the Author):

Review of “The increasing tropical cyclone intensity in the western North Pacific monsoon trough controlled by warming Tibetan Plateau” by Xu et al.

Summary: This study investigated the role of the Tibetan Plateau (TP) in the tropical cyclone (TC) intensity over the western North Pacific (WNP) monsoon trough (MT) area. They found that the recent increase trend of TC intensity in the WNP MT could be primarily attributed to the decreased vertical wind shear (VWS) due to the TP warming, while the contributions from the local/global ocean warming only played a minor role. In addition, the authors postulated that the WNP MT TC intensity would further increase in the future warmer climate.

In general, I find this study to be persuasive; however, there is significant room for improvement in terms of scientific writing and the quality of figures. In addition, I do have reservations about the structure of this manuscript. There are a large number of supplemental figures (15), which is almost five times the total figures included in the main manuscript text (3 main +1 schematic). Moreover, these supplemental figures are referred to in both the results and methodology of this manuscript and seem more integral to the manuscript than just being included as supplemental material.

My comments sorted by major and specific are below. (Please refer to supplementary material for the equations)

Major Comments:

1) Some of the writings are rather confusing. For example, what is “an unstable relationship between TC intensity and sea surface temperature (Line 70)”? In your correlation analysis portion, why you stated, “interdecadal component is the three-year running mean” (Table S1)? The interdecadal timescale refers to a time period or phenomenon that spans multiple decades. It focuses on changes or variations occurring over a period of approximately 10 to 30 years. The three-year temporal window is apparently too short. Similarly, “interannual component is without the linear trend,” this is also wrong. Authors can simply call it detrended correlation.

2) Lots of inconsistencies in the Figure plots, and the quality of the plot are generally poor. For example, the authors used black vectors in Figure 2, but changed them to blue in Figure 3; the “+” sign denotes the area that passed the significance test in the majority of Figures but denotes the area that DID NOT pass the significance test in Figure S15; In Figure 2b, “+” signs are overlaid by the black vectors and very hard to see them clearly. Why did you change to the “x” sign in Figure 2c? I suggest authors replot the Figures made by GrADS with the same standard and better quality.

3) There are a number of instances where this reviewer would have preferred to structurally include the

supplemental figures as regular figures in the text. For example, Supplemental figures 1 and 2 are nice introductory figures about the WNP MT. Why they are considered supplementary when they provide essential background information provided in lines 71-88?

Specific Comments:

1. The description of entropy deficit is inaccurate: Line 122 (and also Lines 431-433) “The energy from the ocean represented by entropy deficit (reflecting the moist static energy from the underlying ocean; see Methods)”. According to Tang and Emanuel (2012), a non-dimensional entropy deficit can be written as:

where S_m^* is the saturation entropy at 600 hPa in the inner core of the TC, S_m is the environmental entropy at 600 hPa, $SSST^*$ is the saturation entropy at the sea surface temperature, and S_b is the entropy of the boundary layer. The numerator is the difference in entropy between the TC and the environment at 600hPa, while the denominator is the air–sea disequilibrium. Therefore, a phrase like “entropy deficit at 600 hPa (Line 431)” and “entropy deficit between ocean and atmosphere (Line 431-432),” is misleading and not appropriate. Do you simply refer to the entropy difference?

Equation (6) is also questionable. Based on Tang and Emanuel (2012), moist entropy can be written as

, why entropy deficit (unit in J/kg/K) can be written into CAPE (unit in J/kg) form? The authors cite Bister and Emanuel (2002) here, but I found no relevant equations in this paper.

2. Lines 339-343: Authors claimed: “Because the CCSM3 model can well capture..., CMIP5 CCSM4 ... are used”. This does not make sense. CCSM4 is the successor to CCSM3, but does it still work well in the East Asian-WNP region? Both CCSM3 and CCSM4 ARE NOT in the Gaussian grids! CAM4 uses finite volume dycore, while Gaussian grids data are typically output from Spectral dycore model like ECMWF IFS.

3. Figure S15d: Panel (d) is apparently very different from panel (c), how can you say “As in (c)”? “temporal curve” is confusing, do you simply mean “timeseries”? Indeed, I am very confused with Figure S15d, is this seasonal mean TC lifetime averaged (or lifetime peak intensity) minimum sea level pressure simulated by RegCM4 during 1986-2050? If so, how did authors identify and track TCs? How many simulated TCs in this area every year? Is it comparable to the observation? Why did you plot TC minimum sea level pressure while the rest of the paper uses 10-m wind speed? A phrase like “minimum sea level pressure (hPa) around the TC center” is very awkward, should be “TC minimum sea level pressure (hPa)”. Line 515 “significant decrease of 1.6 hPa” is too vague. Is this the difference between the 2022-2050 mean and the 1986-2005 mean? Are you sure the 1.6hPa difference is significant?

4. Line 302: “Projections imply a more frequent occurrence of intense typhoons,” did authors find similar results in your RegCM4 downscaling analyses?

5. Line 329: “dotes” should be “dots”.

6. Authors’ definition of OHC is not standard. What is T? “mean temperature in a period” is again too vague. If authors refer the simply OHC, it can be written as

If authors refer to tropical cyclone heat potential (TCHP), it can be written as

7. Line 417-418: Also need to input surface pressure.

8. Line 536: Authors did not use ERA5 data in the analysis, at least did not state it in the paper. Why wrote ERA5 here?

Reviewer #2 (Remarks to the Author):

Recommendation: Accept with revision. The regional TC intensification results linked to TP warming found by the authors are noteworthy and novel, and, if model projections are reliable, may be experienced even more in the future with further TP warming. (Note the caveat of the dependence of these results on the reliability of models for such regional details.) The authors establish that over the period 1988-2018 there is a rising trend in TC intensity in the monsoon trough region of the WNP basin, and a reduction in vertical shear there, which seems to be the important driving factor, since local thermodynamic changes there (SST, PI, etc) are minimal or in the opposite direction needed for intensification. Through correlations and then through some climate model experiments, they show that warming over the Tibetan Plateau region causes a remote atmospheric teleconnection that leads to reduced vertical windshear in the monsoon trough region. They further show that future climate warming scenarios by a climate model have further Tibetan Plateau warming and further reduced vertical wind shear in the MT region. In their methods, they report that a regional dynamical downscaling model based on these climate model simulations produces increased TC intensities in the monsoon trough region, but that effect is very small (only 1.6 hPa decrease in central pressure of TCs over the coming century).

In the manuscript, the authors are not claiming that the change in TC intensity, vertical shear, Tibetan warming, and so forth over 1988-2018 is a detectable anthropogenic change.

See my detailed comments below.

General comment: what caused TP warming (P1 to P2)? It seems that the authors are taking no strong position on this question, but are noting that a further TP warming is projected for the coming century, with some related impacts on VWS in the MT region. Perhaps that should state somewhere that the above question, while important, is outside the scope of this study.

Fig. 1b: why are the TC intensities of the 3 datasets offset from each other?

Line 80-81, does the SAH disappear and an anticyclone appear to the SW or does the anticyclone feature (named SAH) shift to the southwest and get a new different name?

Line 137-138: suggest to say "...TC intensity is more closely linked to VWS than MPI or OHC in the MT region. Therefore, we infer that the JAS weakened VWS during the recent three decades is a dominant factor..."

Line 164-165: based on Ref 37, I suggest to change this slightly to: "This result supports previous findings that WNP TCs develop more slowly when embedded within a monsoon gyre circulation due several mechanisms, including inhibiting effects of vertical wind shear (36,37)."

Line 172: delete "remarkably" here

Line 172-186: I suggest the authors focus more on 200mb winds or vorticity as opposed to a particular geopotential height contour when referring to the eastward shift. With general atmospheric warming throughout the tropical region (e.g., Fig. 2b), the geopotential heights for the 200mb level will be elevated in general (as apparent in Fig. S6e,f for example), making it harder to infer circulation changes based on changes of individual Z200 contours on a map. So they could instead refer to Fig. 2b and Fig. S6B, or to the horizontal gradient features in the Z fields in Fig. Fig. S6 e,f for example to illustrate the eastward shift in circulation associated with the SAH. Individual Z200 contours conflate general large scale warming with more local circulation changes and need to be interpreted with caution.

Line 173-174: "The three-year running mean 600-hPa (close to the surface) geopotential height in the central-western TP (83-88oE, 29o-34oN) has a significant correlation of 0.64 with TC intensity in the MT area (Table S1)..."

This is the first place where the "TPH" index in Table S1 is defined. It needs to be defined in the Table S1 caption. Also the correlation of 0.64 is for interdecadal variations. This seems to be the first place where "interdecadal variations" is defined as three-year running mean. This needs to be made more clear in the text.

Fig. S7c,d should be plotted over the same latitude range to make comparison between them easier.

Line 190-197. I don't find this discussion and interpretation of the results shown in the figures to be entirely convincing. As an example, there is an "A" label for anticyclone on the map for Fig. 2c, but this is a very weak feature. The vertical velocity "wavetrain" from Fig. S8 seems a little more convincing. I recommend deleting some of the weaker material here (lines 194-197).

Line 198-200. Are the authors proposing a mechanism here? This seems to be referring some form of natural internal variability (Asian-Pacific Oscillation). I suggest to either elaborate or delete this sentence.

Line 219: You could say "remarkable" instead of "remarkably"

Lines 219-222: This sentence makes no sense. What does the TC intensity trend have to do with the

removal of a linear trend in TP surface pressure? Rewrite to clarify or delete.

Line 228: "To explore the hypothesis..."

Line 241-242: It is hard to discern the "southward cyclonic anomaly in the lower troposphere" where the "C" is labelled in Fig. 3c. It seems from this and my earlier comment on the "A" feature in Fig. 2c that the authors are trying to stretch to find a correspondence of circulation features between the observations and the TP heating experiments in the MT region and nearby regions. I don't find the resulting discussion to be that convincing on those features.

Line 380-387: Are the authors making the claim that the trends over 1988-2018 are outside of expected natural variability, and if so, what is the basis for this claim? They could test the variations against variations in climate model control runs to see how unusual they are compared to control run variability. (This could not be done for TC intensity but for some of the other environmental changes such as TP warming).

Perhaps the authors are not making any claims about past trends being outside of natural variability. I realize their linkage to anthropogenic climate change is mostly through the use of future projections showing further warming of the TP region and future VWS reduction over the MT region.

Line 483-485: How are the warming Asia and warming Europe experiments constructed?

Line 515-516: A 1.6 hPa decrease in minimum sea level pressure around the TC center in the future scenario (here I assume the authors mean some composite TCs from the future and historical runs) seems like a tiny change for a full century of warming. Is it of any practical importance? Or have I misinterpreted?

END OF REVIEW

Reviewer #3 (Remarks to the Author):

The thesis of this ms. Is that the recent upward trend in strong tropical cyclones in the West Pacific is the result of weakening vertical wind shear, and that in turn is a response to warming of the Tibetan plateau. The argument rests on analyses of observed meteorological data, and numerical simulations. Overall, I would say the case is plausible, if not overwhelmingly convincing. If the argument can be tightened by more careful discussion, the ms. may be suitable for publication.

First, a presentational issue: there are many figures, of which 9 (out of 13 total) are labelled as "supplementary." It is, in fact, quite impossible to follow the main body of the text, containing the key arguments of the paper, without reference to the supplementary figures, leaving one to wonder in what

sense they are “supplementary.” Jumping back and forth between the supposedly important figures and the supplementary ones can be very irritating to the reader (it certainly was to this reviewer).

Scientific issues, in the order they arise:

(line)

(80-81): The SAH migrates south rather than disappears.

(81-83): I have no idea what this sentence is supposed to mean.

(164-165): This sentence is a non sequitur – it is not justified by anything that precedes it in this paragraph.

(177-183): These sentences are not supported by the arguments presented. First, the statement about anomalous upward motion east of the strengthened South Asian High has no basis. It rests on eq. (8) in line 445, which, as the text states, applies only in the cyclonic or anticyclonic centers where vorticity advection is negligible: meaning that it cannot be applied “east of the high.” The rest of the argument, that this leads to increased precipitation, anomalous heating, and a consequent upper level anticyclonic anomaly, is not very convincing.

(203-211): references to “energy” are really inappropriate here. This diagnostic is about wave activity, not energy.

(210-211): This sentence is really a non sequitur. Upward transfer of wave activity (not energy) in no way implies thermal forcing. Any low-level forcing would have the same result.

In general, the results of the model simulations are more convincing than the analysis of observations.

**Reviewer #1 (Remarks to the Author):**

Review of “The increasing tropical cyclone intensity in the western North Pacific monsoon trough
controlled by warming Tibetan Plateau” by Xu et al.

Summary: This study investigated the role of the Tibetan Plateau (TP) in the tropical cyclone (TC)
intensity over the western North Pacific (WNP) monsoon trough (MT) area. They found that the
recent increase trend of TC intensity in the WNP MT could be primarily attributed to the decreased
vertical wind shear (VWS) due to the TP warming, while the contributions from the local/global
ocean warming only played a minor role. In addition, the authors postulated that the WNP MT TC
intensity would further increase in the future warmer climate.

In general, I find this study to be persuasive; however, there is significant room for improvement in
terms of scientific writing and the quality of figures. In addition, I do have reservations about the
structure of this manuscript. There are a large number of supplemental figures (15), which is almost
five times the total figures included in the main manuscript text (3 main +1 schematic). Moreover,
these supplemental figures are referred to in both the results and methodology of this manuscript
and seem more integral to the manuscript than just being included as supplemental material.

**Answer:** We have made substantial revisions to the manuscript, encompassing improvements
in writing, figures, overall structure, and content. We also introduce some statements in Methods to
the text and some of supplementary figures to the regular ones, which reduce the length in Methods
and the number of supplementary figures.

**Major Comments**

**Question 1)** Some of the writings are rather confusing. For example, what is “an unstable
relationship between TC intensity and sea surface temperature (Line 70)”?. In your correlation
analysis portion, why you stated, “interdecadal component is the three-year running mean” (Table
S1)? The interdecadal timescale refers to a time period or phenomenon that spans multiple decades.
It focuses on changes or variations occurring over a period of approximately 10 to 30 years. The
three-year temporal window is apparently too short. Similarly, “interannual component is without
the linear trend,” this is also wrong. Authors can simply call it detrended correlation.

**Answer:** (1) About “what is “an unstable relationship between TC intensity and sea surface
temperature””: Before the cessation of aircraft reconnaissance, the adjusted TC power dissipation
index estimate exhibits a strong correlation with SST. However, post-1988, this correlation weakens
considerably. Moreover, a few studies have reported a high correlation between the Atlantic
Multidecadal Oscillation and TC intensity in the WNP from 1950 to 201829-30. But this strong
correlation has notably diminished after 1987. Thus, the inhomogeneity of TC intensity data due to
the halt in aircraft reconnaissance in 1988 may lead to an unstable relationship between TC intensity
and SST. We have added some statements in **lines 72-81**. More detailed statements are given in
Methods (2).

(2) In line with Li and Chakraborty (2020), who applied a 3-year smoothing technique to TC
 and other time series, we have similarly smoothed all the time series to minimize the impact of
 short-term climatic factors. We have updated the terminology by replacing "interdecadal
 component" with "the smoothed series" and "interannual component" with "detrended component."

**Question 2)** Lots of inconsistencies in the Figure plots, and the quality of the plot are generally poor.
 For example, the authors used black vectors in Figure 2, but changed them to blue in Figure 3; the
 “+” sign denotes the area that passed the significance test in the majority of Figures but denotes the
 area that DID NOT pass the significance test in Figure S15; In Figure 2b, “+” signs are overlaid by
 the black vectors and very hard to see them clearly. Why did you change to the “×” sign in Figure
 2c? I suggest authors replot the Figures made by GrADS with the same standard and better quality.

**Answer:** According the suggestions, we have made improvements to the figures and have
 ensured their consistent presentation throughout the manuscript.

 **Question 3)** There are a number of instances where this reviewer would have preferred to
 structurally include the supplemental figures as regular figures in the text. For example,
 Supplemental figures 1 and 2 are nice introductory figures about the WNP MT. Why they are
 considered supplementary when they provide essential background information provided in lines
 71-88?

**Answer:** We have added some of supplemental figures to the regular ones.

**Specific Comments**

**Question 1.** The description of entropy deficit is inaccurate: Line 122 (and also Lines 431-433)
 “The energy from the ocean represented by entropy deficit (reflecting the moist static energy from
 the underlying ocean; see Methods)”. According to Tang and Emanuel (2012), a non-dimensional
 entropy deficit can be written as:

$$\chi_m = \frac{s_m^* - s_m}{s_{SST}^* - s_b},$$

where s_m^* is the saturation entropy at 600 hPa in the inner core of the TC, s_m is the environmental
 entropy at 600 hPa, s_{SST}^* is the saturation entropy at the sea surface temperature, and s_b is the
 entropy of the boundary layer. The numerator is the difference in entropy between the TC and the
 environment at 600hPa, while the denominator is the air–sea disequilibrium. Therefore, a phrase
 like “entropy deficit at 600 hPa (Line 431)” and “entropy deficit between ocean and atmosphere
 (Line 431-432),” is misleading and not appropriate. Do you simply refer to the entropy difference?

Equation (6) is also questionable. Based on Tang and Emanuel (2012), moist entropy can be written
 as ,

$$s = c_p \log(T) - R_d \log(p_d) + \frac{L_v r_v}{T} - R_v r_v \log(H),$$

why entropy deficit (unit in J/kg/K) can be written into CAPE (unit in J/kg) form? The authors cite
Bister and Emanuel (2002) here, but I found no relevant equations in this paper.

**Answer:** Thank you for your explanation. We now have a better understanding of the
calculations related to entropy deficit. In light of this, we remove the calculation of entropy deficit
from the manuscript because a variable similar to entropy deficit (CAPE*-CAPE) is included in
MPI and this paper considers variables such as MPI, OHC, and SST. This removal of entropy deficit
does not affect our results about the connection between ocean and TC.

**Question 2.** Lines 339-343: Authors claimed: "Because the CCSM3 model can well capture....,
CMIP5 CCSM4 ... are used". This does not make sense. CCSM4 is the successor to CCSM3, but
does it still work well in the East Asian-WNP region? Both CCSM3 and CCSM4 ARE NOT in the
Gaussian grids! CAM4 uses finite volume dycore, while Gaussian grids data are typically output
from Spectral dycore model like ECMWF IFS.

**Answer:** We have removed the statements "Because the CCSM3 model can well capture the
observed atmospheric circulation changes in the East Asian-WNP region (including the tropical
WNP VWS anomaly pattern)" and "Gaussian grids" from the manuscript (lines 350-351).

**Question 3.** Figure S15d: Panel (d) is apparently very different from panel (c), how can you say
"As in (c)"? "temporal curve" is confusing, do you simply mean "timeseries"? Indeed, I am very
confused with Figure S15d, is this seasonal mean TC lifetime averaged (or lifetime peak intensity)
minimum sea level pressure simulated by RegCM4 during 1986-2050? If so, how did authors
identify and track TCs? How many simulated TCs in this area every year? Is it comparable to the
observation? Why did you plot TC minimum sea level pressure while the rest of the paper uses 10-
97 m wind speed? A phrase like "minimum sea level pressure (hPa) around the TC center" is very
awkward, should be "TC minimum sea level pressure (hPa)". Line 515 "significant decrease of 1.6
99 hPa" is too vague. Is this the difference between the 2022-2050 mean and the 1986-2005 mean? Are
100 you sure the 1.6hPa difference is significant?

**Answer:** We have revised the related statements of Fig. S15d in the old version (now in Fig.
S8c). Regarding the identification and tracking of simulated TCs, we employed the TSTORMS
(Detection and Diagnosis of Tropical Storms in High-Resolution Atmospheric Models) software
(www.gfdl.noaa.gov/tstorms/) in our simulations. The detailed statements are seen in lines 496-505.

In our model, the mean annual total number of TCs during 1986-2005 is 9 yr^{-1} , which is less
than that observed (20 yr^{-1}). This model underestimation is likely due to the coarse horizontal
resolution in our simulation, consistent with previous reports by Jin et al. (2016) and Torres-Alavez
et al. (2021). However, since our primary interest is in the changes in TCs, the TC number issue
might become less critical as it affects both present and future cyclones. These related statements
have been added (lines 506-508).

We clarified that the "significant decrease of 1.6 hPa" is the difference between the 2022-2050
 mean and the 1986-2005 mean. This difference is significant at the 98% level. Meanwhile,
 according to your suggestion, we have changed the figure from minimum sea level pressure to
 maximum 10-m wind speed. We now mention the significant increase of 2.4 m/s (by 8.2%) from
 1986-2005 to 2022-2050, which is significant at the 99.9% level (line 324 and Fig. S8c).

**References:**

Jin C S, Cha D H, Lee D K, Suh M S, Hong S Y, Kang H S, and Ho C H 2016. Evaluation of
 climatological tropical cyclone activity over the western North Pacific in the CORDEX-East
 Asia multi-RCM simulations. *Climate Dyn.*, 47, 765–778

Torres-Alavez J A, Glazer R, Giorgi F, Coppola E, Gao X J, Hodges K I, Das S, Ashfaq M, Reale
 121 M, and Sines T 2021. Future projections in tropical cyclone activity over multiple CORDEX
 domains from RegCM4 CORDEX-CORE simulations. *Climate Dyn.*, 57, 1507–1531

**Question 4.** Line 302: “Projections imply a more frequent occurrence of intense typhoons,” did
 authors find similar results in your RegCM4 downscaling analyses?

**Answer:** The similar results are also seen in the RegCM4 downscaling simulations. In Fig. A1,
 there is a notable increase in the occurrence of the intense TC number, suggesting a heightened
 frequency of intense TCs in the future. The associated figure and statements are added (Fig. S8d;
 line 325)

Figure A1. Relationship between the JAS TC number and intensity.

**Question 5.** Line 329: “dotes” should be “dots”.

**Answer:** We have revised.

**Question 6.** Authors’ definition of OHC is not standard. What is $T^{\bar{}}$ “mean temperature in a period”
 is again too vague. If authors refer the simply OHC, it can be written as

$$H = c_p \int_{h_2}^{h_1} \rho(z) T(z) dz$$

If authors refer to tropical cyclone heat potential (TCHP), it can be written as

$$Q = c_p \sum_{Z_0}^{Z_{26}} \rho_i (T_i - 26) \Delta z_i$$

**Answer:** In this revision, according to your suggestion, we have updated the definition of
OHC (Song et al., 2020).

$$143 \quad Q = \sum_{z_0}^{z_{26}} c_{pi} \rho_i (T_i - 26) \Delta z_i$$

Using this revised definition, we obtained consistent results, with no changes in the correlation
between TC and OHC. Please refer to **lines 143, 145, and 409-415 and Table S1**.

**Reference:** Song, J., Duan, Y., and Klotzbach, P., (2020), Increasing trend in rapid intensification
magnitude of tropical cyclones over the western North Pacific, Environmental Research Letters, 15,
084043, <https://doi.org/10.1088/1748-9326/ab9140> .

**Question 7.** Line 417-418: Also need to input surface pressure.

**Answer:** We have added (**line 427**).

**Question 8.** Line 536: Authors did not use ERA5 data in the analysis, at least did not state it in the
paper. Why wrote ERA5 here?

**Answer:** It is due to our mistake. We have deleted it.

**Reviewer #2 (Remarks to the Author):**

Recommendation: Accept with revision. The regional TC intensification results linked to TP
warming found by the authors are noteworthy and novel, and, if model projections are reliable, may
be experienced even more in the future with further TP warming. (Note the caveat of the dependence
of these results on the reliability of models for such regional details.) The authors establish that
over the period 1988-2018 there is a rising trend in TC intensity in the monsoon trough region of
the WNP basin, and a reduction in vertical shear there, which seems to be the important driving
factor, since local thermodynamic changes there (SST, PI, etc) are minimal or in the opposite
direction needed for intensification. Through correlations and then through some climate model
experiments, they show that warming over the Tibetan Plateau region causes a remote atmospheric
teleconnection that leads to reduced vertical windshear in the monsoon trough region. They further
show that future climate warming scenarios by a climate model have further Tibetan Plateau
warming and further reduced vertical wind shear in the MT region. In their methods, they report
that a regional dynamical downscaling model based on these climate model simulations produces
increased TC intensities in the monsoon trough region, but that effect is very small (only 1.6 hPa
decrease in central pressure of TCs over the coming century).

In the manuscript, the authors are not claiming that the change in TC intensity, vertical shear,
Tibetan warming, and so forth over 1988-2018 is a detectable anthropogenic change.

See my detailed comments below.

**General comment:** what caused TP warming (P1 to P2)? It seems that the authors are taking no
strong position on this question, but are noting that a further TP warming is projected for the coming
century, with some related impacts on VWS in the MT region. Perhaps that should state somewhere
that the above question, while important, is outside the scope of this study.

**Answer:** Previous studies have conducted the attribution analysis of the TP warming over the
past decades (covering our study period). It is found that this warming is almost dominated by
anthropogenic influence. The trends of SAT during 1985-2014 in all seasons are positive over the
TP, and the observed warming amplification during 1961-2014 is attributed to anthropogenic
influence. The attributable contribution from anthropogenic influence is estimated to be much larger
than that from natural signal for most warming events in the TP. The associated statements are in
lines 302-312.

**Question 1.** Fig. 1b: why are the TC intensities of the 3 datasets offset from each other?

**Answer:** In order to avoid the confusion, we have modified the presentation of the interquartile
ranges of TC intensity from the three datasets. The previously overlapping ranges, indicated by
shading, have been separated into distinct figures, as shown in Fig. 2b of this revised version.

**Question 2.** Line 80-81, does the SAH disappear and an anticyclone appear to the SW or does the
anticyclone feature (named SAH) shift to the southwest and get a new different name?

**Answer:** We have changed the statements (lines 93-94).

**Question 3.** Line 137-138: suggest to say "...TC intensity is more closely linked to VWS than MPI
or OHC in the MT region. Therefore, we infer that the JAS weakened VWS during the recent three
decades is a dominant factor..."

**Answer:** We have revised (lines 157-159).

**Question 4.** Line 164-165: based on Ref 37, I suggest to change this slightly to: "This result supports
previous findings that WNP TCs develop more slowly when embedded within a monsoon gyre
circulation due several mechanisms, including inhibiting effects of vertical wind shear (36,37)."

**Answer:** According to one reviewer's suggestion, we have deleted the related statement.

**Question 5.** Line 172: delete "remarkably" here

**Answer:** We have deleted.

**Question 6.** Line 172-186: I suggest the authors focus more on 200mb winds or vorticity as opposed
to a particular geopotential height contour when referring to the eastward shift. With general
atmospheric warming throughout the tropical region (e.g., Fig. 2b), the geopotential heights for the
200mb level will be elevated in general (as apparent in Fig. S6e,f for example), making it harder to
infer circulation changes based on changes of individual Z200 contours on a map. So they could
instead refer to Fig. 2b and Fig. S6B, or to the horizontal gradient features in the Z fields in Fig. Fig.
S6 e,f for example to illustrate the eastward shift in circulation associated with the SAH. Individual
Z200 contours conflate general large scale warming with more local circulation changes and need
to be interpreted with caution.

**Answer:** According to your suggestion, we focus on 200 mb winds and consequently remove
the geopotential height contours and the related figure.

**Question 7.** Line 173-174: "The three-year running mean 600-hPa (close to the surface)
geopotential height in the central-western TP (83-88oE, 29o-34oN) has a significant correlation of
0.64 with TC intensity in the MT area (Table S1)..."

This is the first place where the "TPH" index in Table S1 is defined. It needs to be defined in the
Table S1 caption. Also the correlation of 0.64 is for interdecadal variations. This seems to be the
first place where "interdecadal variations" is defined as three-year running mean. This needs to be
made more clear in the text.

**Answer:** We add the related statement and replace 600-hPa (close to the surface) geopotential
height with surface pressure (lines 188-189). This change does not affect the result.

**Question 8.** Fig. S7c,d should be plotted over the same latitude range to make comparison between
them easier.

**Answer:** We have replotted them (see Figs. 4b and 5f).

**Question 9.** Line 190-197. I don't find this discussion and interpretation of the results shown in the
figures to be entirely convincing. As an example, there is an "A" label for anticyclone on the map
for Fig. 2c, but this is a very weak feature. The vertical velocity "wavetrain" from Fig. S8 seems a
little more convincing. I recommend deleting some of the weaker material here (lines 194-197).

**Answer:** According to your suggestion, we have deleted the related statements.

**Question 10.** Line 198-200. Are the authors proposing a mechanism here? This seems to be
referring some form of natural internal variability (Asian-Pacific Oscillation). I suggest to either
elaborate or delete this sentence.

**Answer:** According to your suggestion, we have deleted the related statements.

**Question 11.** Line 219: You could say "remarkable" instead of "remarkably"

**Answer:** We have revised (line 227).

**Question 12.** Lines 219-222: This sentence makes no sense. What does the TC intensity trend have
to do with the removal of a linear trend in TP surface pressure? Rewrite to clarify or delete.

**Answer:** We have deleted the related statements.

**Question 13.** Line 228: "To explore the hypothesis..."

**Answer:** According to your suggestion, we have revised the related statements (line 233).

**Question 14.** Line 241-242: It is hard to discern the "southward cyclonic anomaly in the lower
troposphere" where the "C" is labeled in Fig. 3c. It seems from this and my earlier comment on the
"A" feature in Fig. 2c that the authors are trying to stretch to find a correspondence of circulation
features between the observations and the TP heating experiments in the MT region and nearby
regions. I don't find the resulting discussion to be that convincing on those features.

**Answer:** Following your suggestion for the observation, we also deleted the related statements
for the simulations.

**Question 15.** Line 380-387: Are the authors making the claim that the trends over 1988-2018 are
outside of expected natural variability, and if so, what is the basis for this claim? They could test
the variations against variations in climate model control runs to see how unusual they are compared
to control run variability. (This could not be done for TC intensity but for some of the other
environmental changes such as TP warming).

Perhaps the authors are not making any claims about past trends being outside of natural variability.
I realize their linkage to anthropogenic climate change is mostly through the use of future
projections showing further warming of the TP region and future VWS reduction over the MT
region.

**Answer:** In response to your suggestion, we have analyzed the time series of regional mean
JAS surface temperature over the TP (80-100E, 30-40N) during 100 model years in the
CCSM3_CTL simulation (Fig. A1). This figure illustrates that the TP surface temperature may
exhibit a long-term varying trend. However, previous studies have conducted the attribution of the
TP warming over the past decades (covering our study period). It is found that this warming in the
TP is almost dominated by anthropogenic influence. See our answer to general comment. The
associated statements are in lines 302-312.

Figure A1. The time series of regional mean JAS surface temperature over the TP during 100 model
127 years in the CCSM3_CTL simulation.

**Question 16.** Line 483-485: How are the warming Asia and warming Europe experiments
constructed?

**Answer:** The Asia and Europe experiments are analogous to the TP experiment but focus on
distinct geographical regions (lines 470-471).

**Question 17.** Line 515-516: A 1.6 hPa decrease in minimum sea level pressure around the TC center
in the future scenario (here I assume the authors mean some composite TCs from the future and
historical runs) seems like a tiny change for a full century of warming. Is it of any practical
importance? Or have I misinterpreted?

**Answer:** In this revised version, we accept your suggestion and employ the maximum 10-m
wind to represent TC intensity. This alteration reveals an increase of 2.4 m/s (equivalent to 8.2%)
from the present day to the future, a statistically significant change at the 99.9% level (see line 324
and Fig. S8c).

**Reviewer #3 (Remarks to the Author):**

The thesis of this ms. Is that the recent upward trend in strong tropical cyclones in the West Pacific
is the result of weakening vertical wind shear, and that in turn is a response to warming of the
Tibetan plateau. The argument rests on analyses of observed meteorological data, and numerical
simulations. Overall, I would say the case is plausible, if not overwhelmingly convincing. If the
argument can be tightened by more careful discussion, the ms. may be suitable for publication.

**Question 1.** First, a presentational issue: there are many figures, of which 9 (out of 13 total) are
labelled as “supplementary.” It is, in fact, quite impossible to follow the main body of the text,
containing the key arguments of the paper, without reference to the supplementary figures, leaving
one to wonder in what sense they are “supplementary.“ Jumping back and forth between the
supposedly important figures and the supplementary ones can be very irritating to the reader (it
certainly was to this reviewer).

**Answer:** We have made substantial revisions to the manuscript, introducing some statements
in Methods to the text and some of supplementary figures to the regular ones, which reduce the
length in Methods and the number of supplementary figures.

**Question 2.** (80-81): The SAH migrates south rather than disappears.

**Answer:** We have revised the statement (line 93-94).

**Question 3.** (81-83): I have no idea what this sentence is supposed to mean.

**Answer:** We have deleted the statement “Referring to the similarity of the lower- and upper-
tropospheric circulations in the MT area,”.

**Question 4.** (164-165): This sentence is a non sequitur – it is not justified by anything that precedes
it in this paragraph.

**Answer:** We have deleted the statement.

**Question 5.** (177-183): These sentences are not supported by the arguments presented. First, the
statement about anomalous upward motion east of the strengthened South Asian High has no basis.
It rests on eq. (8) in line 445, which, as the text states, applies only in the cyclonic or anticyclonic
centers where vorticity advection is negligible: meaning that it cannot be applied “east of the high.”
The rest of the argument, that this leads to increased precipitation, anomalous heating, and a
consequent upper level anticyclonic anomaly, is not very convincing.

**Answer:** In this revision, we consider two terms of vorticity equation (not with $w \propto -\beta f^{-1} \frac{\partial v}{\partial z}$,
that is, Eq. (8) of the old version) as follows.

$$w \propto -f^{-1} \frac{\partial}{\partial z} (\mathbf{V} \cdot \nabla \zeta + \beta v)$$

The result shows $\frac{\partial}{\partial z}(\mathbf{V} \cdot \nabla \zeta + \beta v) < 0$ in the region east of the SAH (between 110 and 125E) (Fig.
 A1a), which corresponds to upward motion anomalies (Fig. A1b) and may cause increases in rainfall
 (Fig. 4a) and latent heat of condensation (Fig. A1c). Meanwhile, we also add an analysis on
 temperature advection. Corresponding to a warming TP, tropospheric temperature increases locally,
 westerly wind north of the SAH center might intensify the transport of warmer air toward the east,
 producing positive temperature advection ($-\mathbf{V} \cdot \nabla T$) in the troposphere east of the SAH center (Fig.
 A1d) and increases in temperature in this area. All these may cause the observed eastward extension
 of the warming troposphere, which elongates the tropospheric air column, subsequently raises
 isobaric surfaces in the upper troposphere, and favor the eastward extension of the SAH. Moreover,
 we also compare these observations with the simulations (Figs. S5a-c). They have the good
 consistency though there are northward positions in the simulation. These demonstrate the reliability
 of the observations. The related statements are added in lines 191-202 and 244-246 and the related
 figures are in Figs. S4 and Figs. S5a-c.

Figure A1. (a) Longitude-height section of differences in observed JAS $\frac{\partial}{\partial z}(\mathbf{V} \cdot \nabla \zeta + \beta v)$ ($\times 10^{-15}$
 $\text{m}^{-1}\text{s}^{-1}$) between P2 and P1 along 35°N , in which shaded areas indicate the 95% level. (b) Same as
 in (b) but for p -vertical velocity (Pa s^{-1}). (c) Differences in JAS $\langle Q_2 \rangle$ (W m^{-2}) between P2 and P1.
 (d) Same as in (a) but for temperature advection ($-\mathbf{V} \cdot \nabla T$; K s^{-1}).

**Question 6.** (203-211): references to “energy” are really inappropriate here. This diagnostic is about
 wave activity, not energy.

**Answer:** We have changed to wave “activity”.

**Question 7.** (210-211): This sentence is really a non sequitur. Upward transfer of wave activity (not
 energy) in no way implies thermal forcing. Any low-level forcing would have the same result.

**Answer:** Plumb (1985) pointed out that “there may be a substantial role for diabatic heating in

the forcing of the stationary wave field." Thus, a low-level forcing would yield similar results,
including the low-level dynamic and thermodynamic forcings. Our observed Fz feature (Fig. 4c)
closely resembles the simulation driven by the TP surface heating (Fig. 5e). This similarity suggests
that the upward transfer of wave activity could indeed be caused by thermal forcing.

**Comment:** In general, the results of the model simulations are more convincing than the analysis
of observations.

**Answer:** In fact, we propose the hypothesis that a warming TP affects VWS in the MT area.
We first give the link between TP and VWS and the associated physical processes from observation,
including upper-tropospheric atmospheric circulations, the eastward extensions of both SAH and
temperature from the TP to the western North Pacific, and the meridional propagation of anomalous
wave in the western North Pacific. Especially, we add the analysis for the eastward extension of
SAH anomalies through advection temperature, vertical variation of vorticity, and upward motion.
Then, we rigorously validate this hypothesis and related intermediate processes through comparing
the model results with the observations in detail. Our comparison shows the high similarity between
the simulated and observed physical processes though the simulated positions are systematically
northward. The similarity sufficiently demonstrates our hypothesis.

REVIEWERS' COMMENTS

Reviewer #1 (Remarks to the Author):

I appreciate the authors' effort in addressing my comments. The manuscript has been improved and is now suitable for publication. I only have two minor comments:

1. Line 320: Change "CCSM4" to "CCSM3"
2. Line 79: Reference 29 (Sun, C. et.al. Western tropical Pacific multidecadal variability forced by the Atlantic multidecadal oscillation. Nat. Commun. 8 (1), 1–10(2017)) did not make any analysis about the tropical cyclone. This is not an appropriate citation here.

Reviewer #2 (Remarks to the Author):

Recommendation: Minor revision. See details in the comments below.

Specific Comments.

Line 26: ocean is misspelled

Line 36-38: Replace "is linked to" with: "is very very likely due to "
(it's more than a correlation...it's a causal factor, but it not 100% certain as the causal factor based on the results presented to date)

Line 48: coastal, not costal

Line 48: change "In a warming world" to "In recent decades" unless you demonstrate a physical linkage.

Line 71: Suggest to use "Monsoon Trough" rather than "MT" in the subsection title.

Line 84: "TC activity...is associated..."

Line 86: "beginning in July, reaches its strongest level in August..."

Line 89-90: "...declining below zero again by November..."

Line 103: "Following Ref. 37..." (check journal author guidelines for the exact construction to use)

Line 113: Suggest to say: "This increasing trend is statistically characterized by a change point around 2002..."

Figure 3 caption (line 33). Why does the "(a)" appear in this description of panel (b). Is this a typo? Also could you perhaps start the second sentence as: "(b-e) Differences between P2 and P1 in (b) VWS..." and use the initial few words as the prefix for all of the panels (b-e)? Also rather than "represent the 95% level" you could say "represent statistically significant results at the $p=0.05$ level" and later on say "represent the $p=0.05$ level" (2 occurrences)

Throughout manuscript:

Change nomenclature from 95, 99, 99.9%, etc. level to significant (or statistically significant) at the $p=0.05$, $p=0.01$, or $p=0.001$ level.

(See for example: https://en.wikipedia.org/wiki/Statistical_significance or <https://www.nlm.nih.gov/oet/ed/stats/02-930.html>)

Line 163: Could you say "anticlockwise vertical windshear anomaly pattern" rather than "anticlockwise circulation pattern"?

Line 222: "extends" not "expands"

Line 224: "increases" not "experiences an increases"

Line 225: "(significant at the 99.9% level)" or even better to say "(significant at the 0.001 level)"

Line 255: Suggest to say: "...captures a response similar to the observed changes..."

Line 261: "Our results strongly suggest that..." (Demonstrates is too strong a term here).

Line 267-268: "A significant warming" (not "The significant warming" and "we conduct sensitivity ..."
(drop "the")

Line 281: "One might speculate that global pattern of SST change from P1 to P2 plays the dominant role in the weakened..." (this is needed to clarify that you are not performing an experiment with a globally uniform change of SST).

Line 284: "...results show that the observed global distribution of SST change..."

Line 292: "These studies have demonstrated the modeled impact of..."

Line 295: "Our results strongly suggest that relative to a..."

Line 307-308: "...(CMIP6), previously studies consistently find that the recently accelerated warming in the TP is very likely dominated by anthropogenic..."

Line 310: "...during 1961-2014 has been attributed to ..."

Line 324-325: "...8.2%, significant at the 0.001 level)...notable increase in the number of intense TCs (Fig. S8c-d)..."

Line 325-326: "...suggest a more frequent occurrence of intense typhoons in the region in the coming decades, which heightens..."

Line 388: "...revealing a significant change-point in the timeseries around 2002..."

Line 408-429: Although it is not likely to change any conclusions of this study (owing to the small influence of SST on TC intensity in the region relative to vertical shear), you could mention here and in the main paper the work of I-I Lin and collaborators looking at a version of MPI that includes ocean temperature averaged down to a certain depth (and their work on vertical profile of ocean temperature change and its influence on TC intensity.)

Reviewer #1

Minor comments:

Question 1. Line 320: Change “CCSM4” to “CCSM3”

Answer: We have changed (line 321).

Question 2. Line 79: Reference 29 (Sun, C. et al. Western tropical Pacific multidecadal variability forced by the Atlantic multidecadal oscillation. Nat. Commun. 8 (1), 1-10(2017)) did not make any analysis about the tropical cyclone. This is not an appropriate citation here.

Answer: We have deleted this reference and adjusted the order of the related references.

Thank you very much.

Reviewer #2

Minor comments:

Question 1. Line 26: ocean is misspelled

Answer: We have changed (line 26).

Question 2. Line 36-38: Replace “is linked to” with: “is very likely due to” (it’s more than a correlation ... it’s a causal factor, but it not 100% certain as the causal factor based on the results presented to date)

Answer: We have changed (line 33). Because the abstract words are largely reduced and the final paragraph in introduction is rewritten, the line number of the manuscript changes.

Question 3. Line 48: coastal, not costal

Answer: We have changed (line 39).

Question 4. Line 48: change “In a warming world” to “In recent decades” unless you demonstrate a physical linkage.

Answer: We have changed (line 39).

Question 5. Line 71: Suggest to use “Monsoon trough” rather than “MT” in the subsection title.

Answer: We have changed (line 68).

Question 6. Line 84: “TC activity ... is associated ...”

Answer: We have changed (line 81).

Question 7. Line 86: “beginning in July, reaches its strongest level in August...”

Answer: We have changed (line 82-83).

Question 8. Line 89-90: “... declining below zero again by November...”

Answer: We have changed (line 86).

Question 9. Line 103: “Following Ref. 37...” (check journal author guidelines for the exact construction to use)

Answer: We have changed to “Following Ref. 36, ...” (line 99). Because a reference in the old version is deleted according to another reviewer’s suggestion, “Ref. 37” of the old version is changed to “Ref. 36”.

Question 10. Line 113: Suggest to say: “This increasing trend is statistically characterized by a change point around 2002...”

Answer: We have changed (line 110).

Question 11. Figure 3 caption (line 33). Why does the “(a)” appear in this description of panel (b). Is this a typo? Also could you perhaps start the second sentence as: “(b-e)” Differences between P2 and P1 in (b) VWS...” and use the initial few words as the prefix for all of the panels (b-e)? Also rather than “represent the 95% level” you could say “represent statistically significant results at the $p=0.05$ level” and later on say “represent the $p=0.05$ level” (2 occurrences)

Answer: Yes. It is due to a typo. We have changed the caption of this figure according to your suggestions.

Question 12. Throughout manuscript: Change nomenclature from 95, 99, 99.9%, etc. level to significant (or statistically significant) at the $p=0.05$, $p=0.01$, or $p=0.001$ level. (see for example: https://en.wikipedia.org/wiki/Statistical_significance or <https://WWW.nlm.nih.gov/oet/ed/stats/02-930.html>)

Answer: We have changed them throughout manuscript (marked by red).

Question 13. Line 163: Could you say “anticlockwise vertical windshear anomaly patten” rather than “anticlockwise circulation pattern”

Answer: We have changed (line 161-162).

Question 14. Line 222: “extends” not “expand”

Answer: We have changed (line 222).

Question 15. Line 224: “increases” not “experiences an increase”

Answer: We have changed (line 223).

Question 16. Line 225: “(significant at the 99.9% level)” or even better to say “(significant at the 0.001 level)”

Answer: We have changed (line 225).

Question 17. Line 255: Suggest to say “... capture a response similar to the observed changes”

Answer: We have changed (line 256).

Question 18. Line 261: “Our results strongly suggest that ...” (demonstrates is too strong a term here).

Answer: We have changed (line 262).

Question 19. Line 267-268: “A significant warming” (not “The significant warming” and “we conduct sensitivity ...” (drop “the”)

Answer: We have changed (line 268-269).

Question 20. Line 281: “One might speculate the global pattern of SST change from P1 to P2 plays the dominant role in the weakened...” (this is needed to clarify that you are not performing an experiment with a globally uniform change of SST)

Answer: We have changed (line 283-284).

Question 21. Line 284: “... results show that the observed global distribution of SST change ...”

Answer: We have changed (line 286).

Question 22. Line 292: “These studies have demonstrated the modeled impact of ...”

Answer: We have changed (line 295).

Question 23. Line 295: “Our results strongly suggest that relative to a ...”

Answer: We have changed (line 297).

Question 24. Line 307-307: “... (CMIP6), previously studies consistently find that the recently accelerated warming in the TP is very likely dominated by anthropogenic...”

Answer: We have changed (line 308-309).

Question 25. Line 310: “... during 1961-2014 has been attributed to ...”

Answer: We have changed (line 311).

Question 26. Line 324-325: “... 8.2%, significant at the 0.001 level)... notable increase in the number of intense TCs (Fig. S8c-d)...”

Answer: We have changed (line 326-327).

Question 27. Line 325-326: “... suggest a more frequent occurrence of intense typhoons in the region in the coming decades, which heights...”

Answer: We have changed (line 328).

Question 28. Line 388: “... revealing a significant change-point in the timeseries around 2022”

Answer: We have changed (line 390).

Question 29. Line 408-429: Although it is not likely to change any conclusions of this study (owing to the small influence of SST on TC intensity in the region relative to vertical shear), you could mention here and in the main paper the work of I-I Lin and collaborators looking at a version of MPI that includes ocean temperature averaged down to a certain depth (and their work on vertical profile of ocean temperature change and its influence on TC intensity.)

Answer: We have added some statements about the work of I-I Lin in the main paper and the method part (line 139 and line 431-432).

Thank you very much.